# Health benefits attributed to 17α-estradiol, a lifespan-extending compound, are mediated through estrogen receptor α

Shivani N Mann[1,2,3], Niran Hadad[4], Molly Nelson Holte[5], Alicia R Rothman[1], Roshini Sathiaseelan[1], Samim Ali Mondal[1], Martin-Paul Agbaga[3,6,7], Archana Unnikrishnan[3,8], Malayannan Subramaniam[5], John Hawse[5], Derek M Huffman[9], Willard M Freeman[2,10,11], Michael B Stout[1,2,3]*

[1]Department of Nutritional Sciences, University of Oklahoma Health Sciences Center, Oklahoma City, United States; [2]Oklahoma Center for Geroscience, University of Oklahoma Health Sciences Center, Oklahoma City, United States; [3]Harold Hamm Diabetes Center, University of Oklahoma Health Sciences Center, Oklahoma City, United States; [4]The Jackson Laboratory, Bar Harbor, United States; [5]Department of Biochemistry and Molecular Biology, Mayo Clinic, Rochester, United States; [6]Department of Cell Biology, University of Oklahoma Health Sciences Center, Oklahoma City, United States; [7]Dean McGee Eye Institute, University of Oklahoma Health Sciences Center, Oklahoma City, United States; [8]Department of Biochemistry and Molecular Biology, University of Oklahoma Health Sciences Center, Oklahoma City, United States; [9]Department of Molecular Pharmacology, Albert Einstein College of Medicine, New York, United States; [10]Genes & Human Disease Research Program, Oklahoma Medical Research Foundation, Oklahoma City, United States; [11]Oklahoma City Veterans Affairs Medical Center, Oklahoma City, United States

*For correspondence:
michael-stout@ouhsc.edu

Competing interests: The authors declare that no competing interests exist.

**Abstract** Metabolic dysfunction underlies several chronic diseases, many of which are exacerbated by obesity. Dietary interventions can reverse metabolic declines and slow aging, although compliance issues remain paramount. 17α-estradiol treatment improves metabolic parameters and slows aging in male mice. The mechanisms by which 17α-estradiol elicits these benefits remain unresolved. Herein, we show that 17α-estradiol elicits similar genomic binding and transcriptional activation through estrogen receptor α (ERα) to that of 17β-estradiol. In addition, we show that the ablation of ERα completely attenuates the beneficial metabolic effects of 17α-E2 in male mice. Our findings suggest that 17α-E2 may act through the liver and hypothalamus to improve metabolic parameters in male mice. Lastly, we also determined that 17α-E2 improves metabolic parameters in male rats, thereby proving that the beneficial effects of 17α-E2 are not limited to mice. Collectively, these studies suggest ERα may be a drug target for mitigating chronic diseases in male mammals.

## Introduction

Aging is the leading risk factor for most chronic diseases, many of which are associated with declines in metabolic homeostasis (López-Otín et al., 2013). Metabolic detriments associated with advancing age are further exacerbated by obesity (Villareal et al., 2005; Waters et al., 2013), which has risen

substantially in the older population (>65 years) over the past several decades (*Flegal et al., 2010*; *Flegal et al., 2016*). Moreover, obesity in mid-life has been shown to accelerate aging mechanisms and induce phenotypes more commonly observed in older mammals (*Bischof and Park, 2015*; *Horvath et al., 2014*; *Nevalainen et al., 2017*; *Yang et al., 2009*; *Whitmer et al., 2005a*; *Whitmer et al., 2005b*; *Dye et al., 2017*). These observations have led many to postulate that obesity may represent a mild progeria syndrome (*Salvestrini et al., 2019*; *Tzanetakou et al., 2012*; *Pérez et al., 2016*; *Tchkonia et al., 2010*; *Stout et al., 2017a*). Although it is well established that dietary interventions, including calorie restriction, can reverse obesity-related metabolic sequelae, many of these strategies are not well tolerated in older patients due to concomitant comorbidities (*Villareal et al., 2005*; *Jensen et al., 2014*). Compliance issues across all age groups also remain a paramount hurdle due to calorie restriction adversely affecting mood, thermoregulation, and musculoskeletal mass (*Dirks and Leeuwenburgh, 2006*). These adverse health outcomes demonstrate the need for pharmacological approaches aimed at curtailing metabolic perturbations associated with obesity and aging.

17α-estradiol (17α-E2) is one of the more recently studied compounds to demonstrate efficacy for beneficially modulating obesity- and age-related health outcomes. The NIA Interventions Testing Program (ITP) found that long-term administration of 17α-E2 extends median lifespan of male mice in a dose-dependent manner (*Strong et al., 2016*; *Harrison et al., 2014*). Our group has been exploring potential mechanisms by which 17α-E2 may improve healthspan and extend lifespan in a sex-specific manner. We have found that 17α-E2 administration reduces calorie intake and regional adiposity in combination with significant improvements in a multitude of systemic metabolic parameters in both middle-aged obese and old male mice without inducing deleterious effects (*Stout et al., 2017b*; *Steyn et al., 2018*; *Miller, 2020*). Other groups have also determined that lifelong administration of 17α-E2 beneficially modulates metabolic outcomes, including glucose tolerance, mTORC2 signaling, and hepatic amino acid composition and markers of urea cycling, which were reported to be dependent upon the presence of endogenous androgens (*Garratt et al., 2017*; *Garratt et al., 2018*). Additionally, multiple lifespan extending compounds, including 17α-E2, exhibit similar modifications in liver function (*Tyshkovskiy et al., 2019*). In all, recent studies by several independent laboratories strongly indicate that the lifespan-extending effects of 17α-E2 are at least associated with, if not dependent on, metabolic improvements.

Despite the mounting evidence demonstrating that 17α-E2 improves numerous health parameters, the signaling mechanism(s) and primary tissues through which 17α-E2 elicits these benefits remain unknown. Although 17α-E2 is a naturally occurring enantiomer to 17β-estradiol (17β-E2), it has been postulated that 17α-E2 signals through a novel uncharacterized receptor (*Toran-Allerand, 2005*; *Toran-Allerand et al., 2002*; *Toran-Allerand et al., 2005*; *Green and Simpkins, 2000*) as opposed to classical estrogen receptors α (ERα) and β (ERβ), which is due to 17α-E2 having significantly reduced binding affinity for ERα and ERβ as compared to 17β-E2 (*Edwards and McGUIRE, 1980*; *Korenman, 1969*; *Littlefield et al., 1990*; *Anstead et al., 1997*). For this reason, 17α-E2 is often referred to as a non-feminizing estrogen (*Green and Simpkins, 2000*; *Engler-Chiurazzi et al., 2017*; *Kaur et al., 2015*). A few studies have suggested that a novel but uncharacterized estrogen receptor, termed ER-X, may mediate 17α-E2 actions in the brain (*Toran-Allerand, 2005*; *Toran-Allerand et al., 2002*; *Toran-Allerand et al., 2005*; *Green and Simpkins, 2000*), although more recent studies supporting this hypothesis are lacking in the literature. Similarly, no reports to date have directly tested whether the doses of 17α-E2 shown to improve healthspan and lifespan in mice are mediated through ERα and/or ERβ.

There is a multitude of data in the diabetes and metabolism literature demonstrating that ERα is a regulator of systemic metabolic parameters. Although most of these studies have historically been performed in female mammals, more recent studies have demonstrated that ERα also plays a critical role in modulating metabolism in male mammals. For instance, Allard and colleagues recently demonstrated that genomic actions of ERα regulate systemic glucose homeostasis in mice of both sexes and insulin production and release in males (*Allard et al., 2019*). Other studies have also determined that hepatic steatosis and insulin sensitivity, and therefore the control of gluconeogenesis, are regulated through FOXO1 in an ERα-dependent manner in male mice (*Yan et al., 2019*). Furthermore, hepatocyte-specific deletion of ERα was sufficient to abrogate similar estrogen-mediated metabolic benefits (*Guillaume et al., 2019*; *Qiu et al., 2017*; *Meda et al., 2020*). Given that several reports have linked the administration of 17α-E2 to improvements in metabolic homeostasis, we

hypothesized that 17α-E2 signals through ERα to modulate hepatic function and systemic metabolism, thereby potentially contributing to the lifespan-extending effects of 17α-E2.

The work outlined in this report sought to determine if ERα is the primary receptor by 17α-E2 signals and modulates health parameters in mice. We initially determined that 17α-E2 and 17β-E2 elicit similar genomic actions through ERα. Given that no studies to date have tested the potential role of ERα in modulating 17α-E2-mediated effects in vivo, we treated obese wild type (WT) and ERα knockout (ERα KO) littermate mice with 17α-E2 to determine if the ablation of ERα could attenuate 17α-E2-induced benefits on adiposity, metabolic homeostasis, and hepatic function. We found that the ablation of ERα completely attenuated all beneficial metabolic effects of 17α-E2. Follow-up studies in male WT rats undergoing hyperinsulinemic-euglycemic clamps revealed that 17α-E2 modulates hepatic insulin sensitivity following acute exposure. Given the established connection between the hypothalamus and liver in the modulation of hepatic insulin sensitivity (*Könner et al., 2007*; *Ruud et al., 2017*; *Pocai et al., 2005a*; *Pocai et al., 2005b*; *Dodd et al., 2018*), coupled with our data demonstrating ERα-dependency of 17α-E2 actions on metabolic parameters, we speculate that 17α-E2 acts through ERα in the liver and/or hypothalamus to improve metabolic homeostasis in male mammals.

## Results

### 17α-E2 and 17β-E2 similarly modulate genomic binding and transcriptional activity of ERα

Ligand-mediated ERα dimerization leads to nuclear translocation and transcriptional activity. Previous work has shown that 17α-E2 and 17β-E2 can bind to ERα with different affinities (*Edwards and McGUIRE, 1980*; *Korenman, 1969*; *Littlefield et al., 1990*; *Anstead et al., 1997*), yet potential differences in resultant genomic binding and transcriptional activity between the two ligands remains unexplored. We assessed ERα DNA binding and transcriptional induction following exposure to 17β-E2 (10 nM) or 17α-E2 (10 nM or 100 nM) in U2OS cells that stably express ERα following doxycycline induction. We chose to use these cells because they do not endogenously express any form of ERα or ERβ and have been extensively utilized to elucidate the effects of ERα and ERβ agonists and antagonists on gene expression (*Monroe et al., 2003*; *Monroe et al., 2005*). ChIP-sequencing revealed peaks of ERα genomic binding in all conditions, that when compared, are qualitatively similar across treatments (*Figure 1A*). Statistically significant differences in ERα binding were determined by negative binomial regression with a Wald's pairwise post-hoc comparison (false discovery rate correction, FDR < 0.05). A total of 21,443 peaks were found to have a significant pairwise post-hoc comparison between vehicle and 17α-E2 and/or 17β-E2 treated cells. No statistically significant differences between 17α-E2 and 17β-E2-treated groups were observed. 17α-E2 and 17β-E2 not only induced ERα binding at the same genomic locations but also to similar magnitudes. Comparing the levels of increased or decreased ERα binding (as compared to vehicle control) between treatments demonstrates the consistency of ERα genomic binding regardless of the agonist (*Figure 1B*, *Supplementary file 1*). The degree of increased or decreased ERα binding was highly similar between 10 nM 17α-E2 and 10 nM 17β-E2, (Pearson's r = 0.95, p<0.001) and 100 nM 17α-E2 and 10 nM 17β-E2 (Pearson's r = 0.96, p<0.001). As expected, ERα-binding sites were enriched for estrogen response elements (ERE), estrogen-related receptor beta (Esrrb), and estrogen-related receptor alpha (Erra). Other common motifs found within ER elements, including steroidogenic factor-1 (SF1) (*Lin et al., 2007*), and motif elements of known interacting partners, including retinoid acid receptor:retinoid X receptor (RAR:RXR) (*Lee et al., 1998*), were also enriched (*Figure 1C*). In addition, we observed enrichment of androgen response elements (ARE) in ERα peaks (*Figure 1C*). Of particular relevance, many of the top enriched motifs identified contained the ERE consensus sequence TTGAC (*Supplementary file 2*). Following motif enrichment, we performed pairwise differential motif enrichment across all groups to determine if a specific agonist or agonist concentration caused a differential enrichment of any motifs, as would be suggestive of differential genomic binding. No differential motif binding was observed across treatment groups indicating that both 17α-E2 and 17β-E2 cause ERα to bind to the same types of genomic elements (Hypergeometric test, FDR < 0.05).

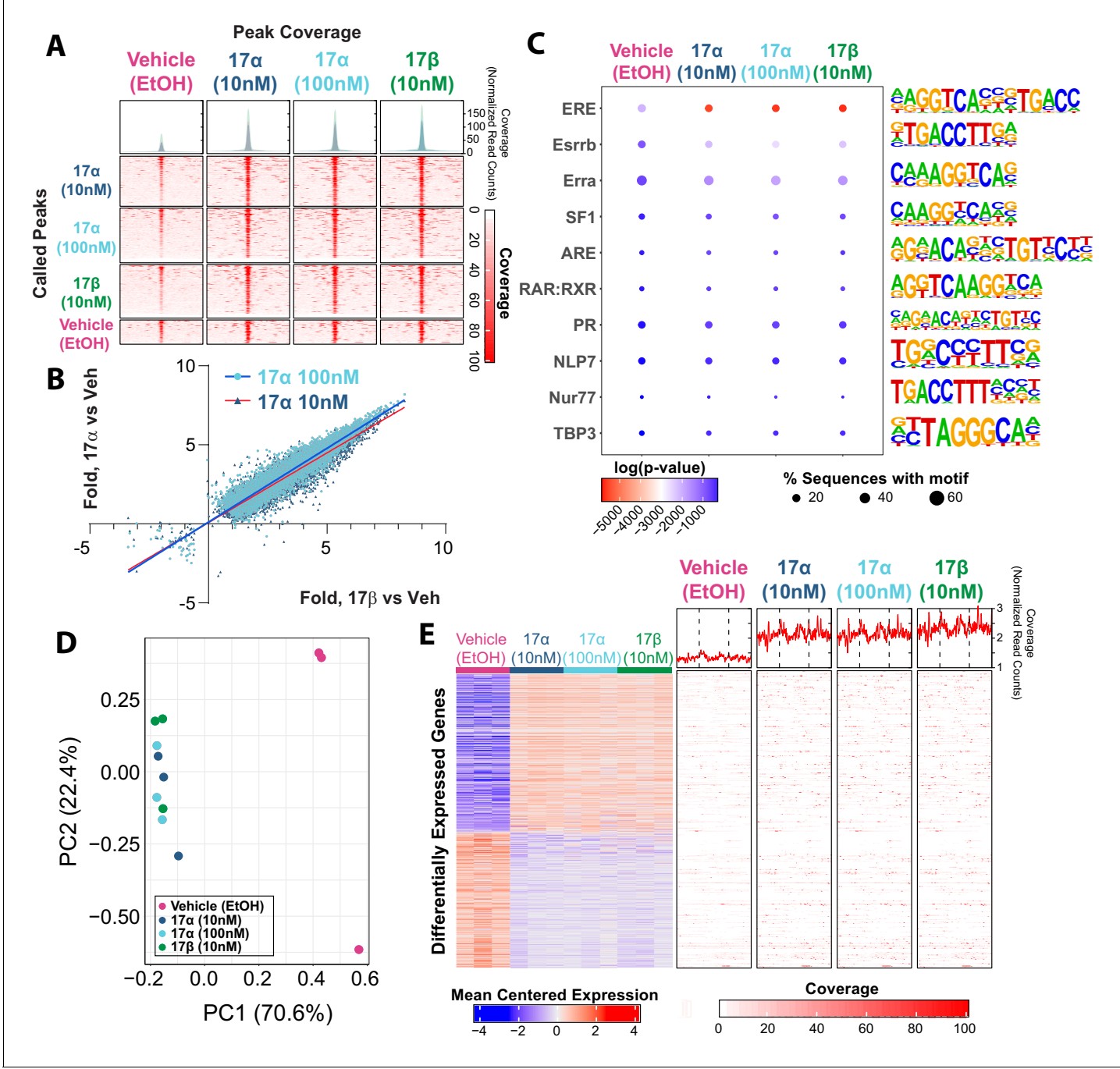

**Figure 1.** 17α-E2 and 17β-E2 elicit similar genomic binding and transcriptional profiles through ERα. (A) Heatmap representing normalized genome-wide DNA binding by ERα via ChIP sequencing analyses centered according to peak summits for each treatment group and compared to each other group. (B) Differential binding was identified between vehicle and 17α-E2 or 17β-E2 treatment groups, but no differences were identified between 17α-E2 and 17β-E2-treated groups (negative binomial regression, followed by Wald test for pairwise comparisons, FDR < 0.05). Fold change in binding relative to vehicle control was compared between 17α-E2 treatments and 17β-E2. (C) Motif enrichment analysis, filtered for mammalian and non-overlapping motif groups, showing the top 10 non-redundant enriched sequence motifs across treatment groups (hypergeometric test, FDR < 0.05), (D) PCA plot of transcriptional profiles by RNA sequencing analyses demonstrating clustering of 17α-E2 and 17β-E2 treatment groups together, opposite from Vehicle-treated group along the first principle component, and (E) Heatmap representing differentially expressed genes (negative binomial regression, followed by Wald test for pairwise comparisons, FDR < 0.05) by RNA sequencing analyses (left) and ERα binding patterns within the gene body ±5 kb flanking regions of these genes via ChIP sequencing (right). Significant differential pairwise expression was observed only between 17α-E2 and 17β-E2 treatment groups and vehicle-treated control. These studies utilized U2OS-ERα cells treated with low dose (10 nM) 17α-E2, high dose (100 nM) 17α-E2, 17β-E2 (10 nM), or vehicle (EtOH). n = 3/group.

*Figure 1 continued on next page*

*Figure 1 continued*

The online version of this article includes the following figure supplement(s) for figure 1:

**Figure supplement 1.** 17α-E2 and 17β-E2 elicit similar ERα binding profile.

Next, we examined potential differences in transcriptional responses between treatment groups using RNA-sequencing. Principle component analysis based on the entire transcriptome revealed that all samples exposed to either 17α-E2 or 17β-E2 clustered together, whereas vehicle-treated cells remained distinctly separated from treated cells on the first principle component, which explains the majority of the variance in transcription (70.6%) (*Figure 1D*). These data suggest that treatment vs vehicle is the primary covariate explaining variance in transcriptional profiles, not the specific agonist. Next, differential expression was assessed between all groups using a negative binomial regression model with a Wald pairwise post-hoc test. No genes were found to be differentially regulated (FDR < 0.05) between the estrogen treatments. Yet, compared to vehicle-treated cells, treatment of U2OS cells with either 10 nM or 100 nM 17α-E2 or 10 nM 17β-E2 resulted in nearly identical gene suppression and activation signatures (*Figure 1E*, left). Additionally, both 17α-E2 or 17β-E2 treatment conditions resulted in higher ERα DNA binding affinity to gene bodies of these differentially expressed transcripts compared to vehicle treatment, and no differences were observed between 17α-E2 and 17β-E2 conditions (*Figure 1E*, right) (negative binomial regression with Wald pairwise post-hoc). These findings led us to postulate that 17α-E2 may be the signaling through ERα to modulate health parameters in male mice. As such, we subsequently sought to determine if the ablation of ERα in vivo would mitigate the effects of 17α-E2.

## ERα ablation attenuates 17α-E2-mediated benefits on metabolic parameters in male mice in vivo

To induce obesity and metabolic perturbations in male mice, we administered high-fat diet (HFD) for several months prior to initiating 17α-E2 treatment. Control mice remained on HFD, whereas 17α-E2-treated mice were switched to an identical HFD containing 17α-E2. Almost immediately after 17α-E2 treatment began, male WT mice displayed significant reductions in mass (*Figure 2A–B*) and adiposity (*Figure 2C–D*). This is aligned with our previous reports demonstrating that 17α-E2 administration quickly reduces body mass and adiposity (*Stout et al., 2017b*; *Steyn et al., 2018*; *Miller, 2020*), which we have linked to hypothalamic regulation of anorexigenic signaling pathways (*Steyn et al., 2018*). Indeed, male WT mice in the current study also displayed robust declines in calorie consumption during the first 4 weeks of treatment (*Figure 2E*). Conversely, all these benefits were completely abolished in male mice lacking ERα (ERα KO), thereby confirming that 17α-E2 definitively acts through ERα to modulate feeding behaviors, mass, and adiposity in male mice. Given the close association between adiposity and metabolic homeostasis, coupled with our previous work demonstrating the ability of 17α-E2 to improve metabolic parameters (*Stout et al., 2017b*; *Steyn et al., 2018*), we also assessed several metabolic variables in these studies. Similar to the mass and adiposity data described above, male WT mice receiving 17α-E2 displayed significant improvements in fasting insulin (*Figure 3B*), HbA1C (*Figure 3C*), and glucose tolerance (*Figure 3D–E*, *Figure 3—figure supplement 1*), whereas male ERα KO mice receiving 17α-E2 failed to recapitulate these findings. Interestingly, despite the masses of the male WT 17α-E2 treatment group being nearly 15 grams greater than those of the male WT chow-fed controls, glucose tolerance was essentially identical between these groups, thereby indicating that 17α-E2 restores metabolic flexibility in the presence of obesity in male mice (*Figure 3D–E*, *Figure 3—figure supplement 1*). We also evaluated the effects of 17α-E2 on metabolic parameters in female WT and ERα KO mice provided a standard chow diet. In contrast to the males, we chose not to subject female WT and ERα KO mice to HFD because female ERα KO mice spontaneously develop obesity due to the ablation of ERα (*Manrique et al., 2012*; *Vidal et al., 1999*). Given that the female ERα KO mice are already in a challenged state, HFD would further exacerbate mass and adiposity differences between ERα KO and WT female mice. We found that 17α-E2 failed to elicit improvements in mass, adiposity, calorie consumption, or metabolic parameters in female mice of either genotype (*Figure 3—figure supplement 2*). The positive effects of 17α-E2 in male mice led us to speculate that the liver may play a key role in modulating 17α-E2-mediated effects on systemic metabolic homeostasis. Importantly, several

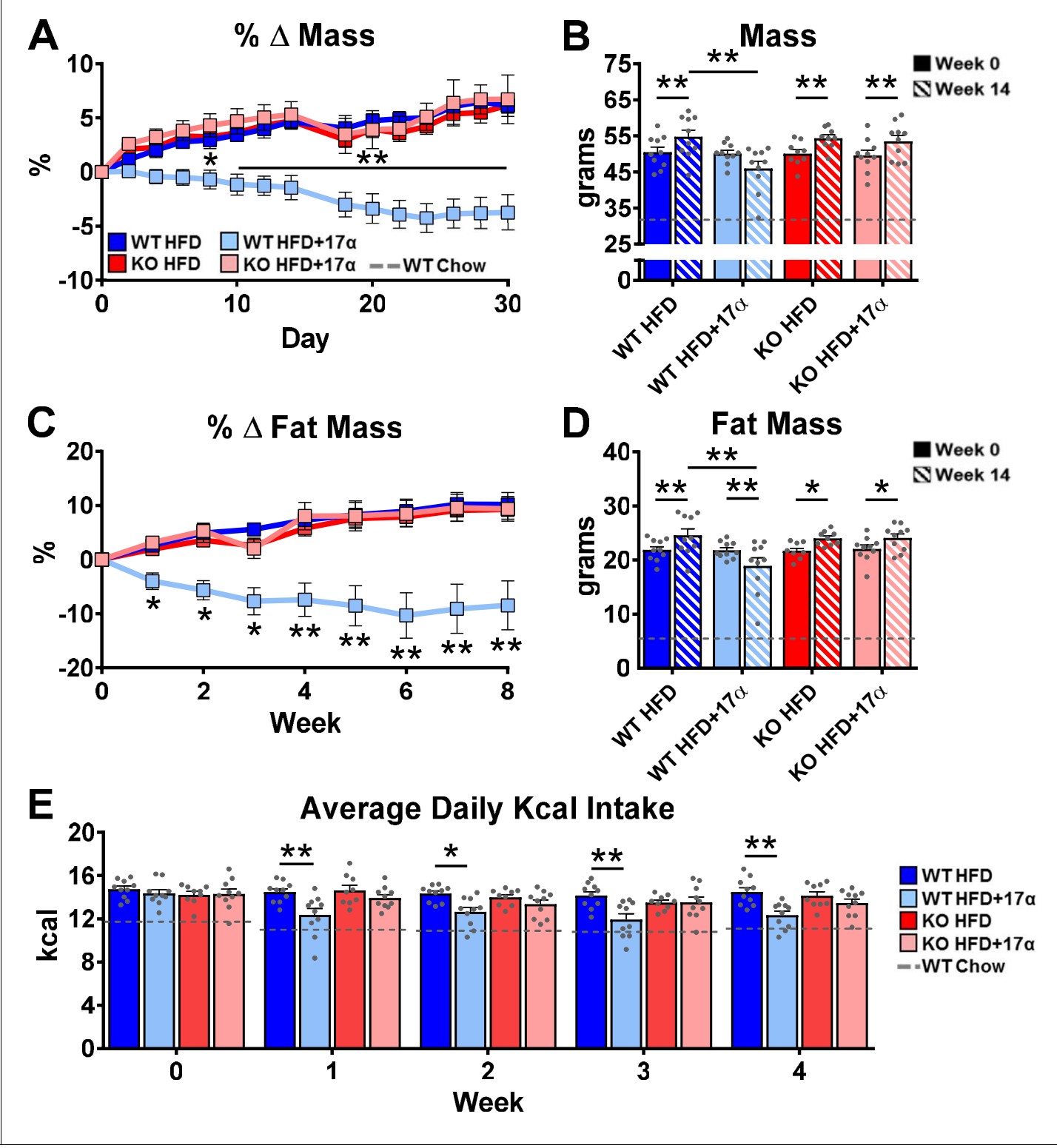

**Figure 2.** ERα is required for 17α-E2 to reduce mass, adiposity, and calorie intake in male mice. (A) Percent change in mass (mean ± SEM, two-way repeated measures ANOVA with Holm-Sidak post-hoc; *p<0.05, **p<0.005 between WT HFD and WT HFD+17α), (B) Mass at baseline (week 0; solid) and week 14 (striped) (mean ± SEM, two-way repeated measures ANOVA with Holm-Sidak post-hoc; *p<0.05, **p<0.005), (C) Percent change in fat mass (mean ± SEM, two-way repeated measures ANOVA with Holm-Sidak post-hoc; *p<0.05, **p<0.005), (D) Fat mass at baseline (week 0; solid) and week 14 (striped) (mean ± SEM, two-way repeated measures ANOVA with Holm-Sidak post-hoc; *p<0.05, **p<0.005), and (E) Average daily calorie intake per week in WT and ERα KO mice provided 45% HFD (TestDiet 58V8)±17α-E2 (14.4ppm) (mean ± SEM, two-way repeated measures ANOVA with

*Figure 2 continued on next page*

studies have implicated hepatic ERα in the regulation of glucose homeostasis, insulin sensitivity, and crosstalk with hypothalamic neurons that modulate metabolism and feeding behavior (*Meda et al., 2020*; *Torre et al., 2017*; *Barros and Gustafsson, 2011*).

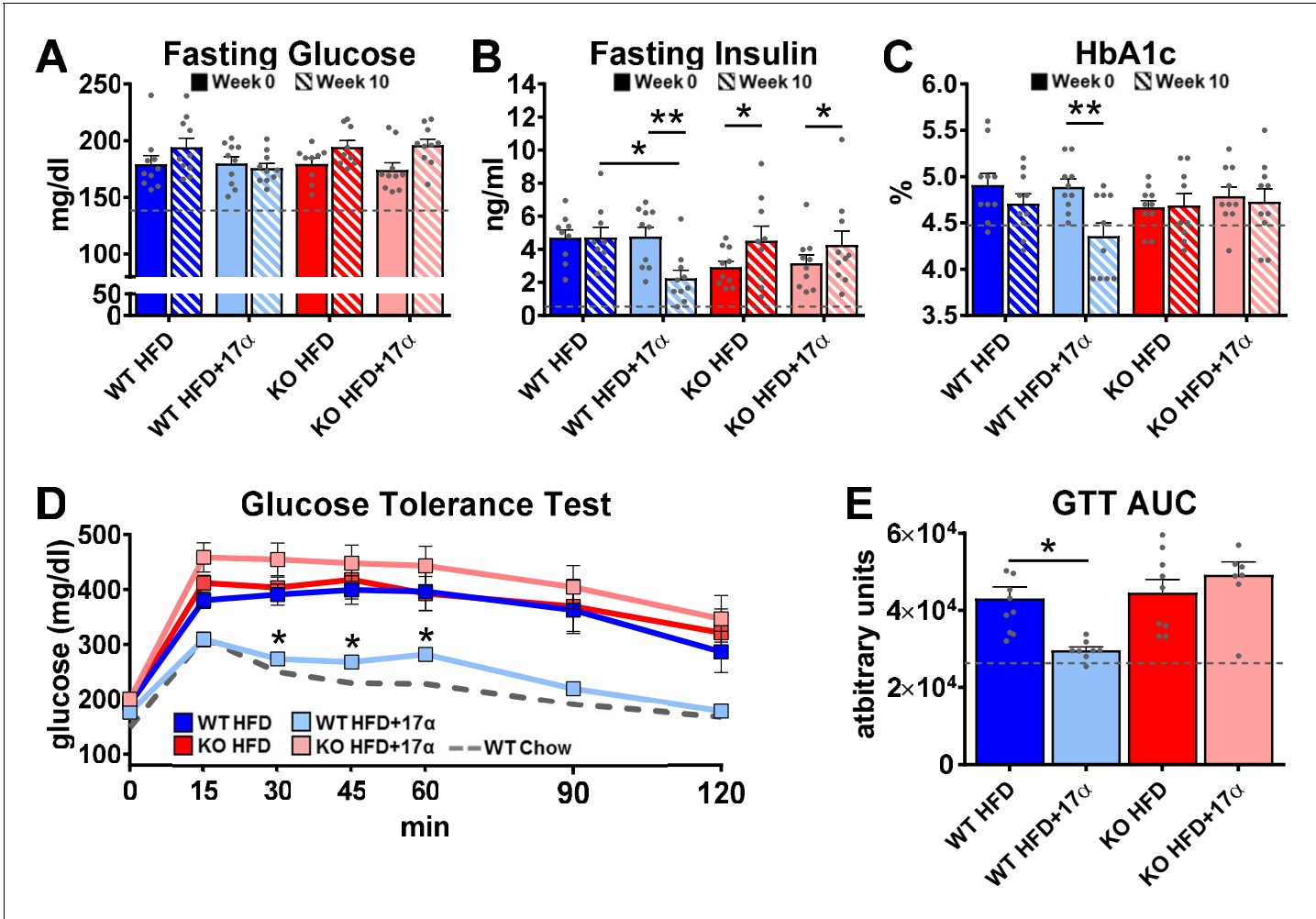

**Figure 3.** 17α-E2 reverses obesity-related metabolic dysfunction in male WT, but not ERα KO, mice. (**A**) Fasting glucose (mean ± SEM, two-way repeated measures ANOVA), (**B**) Fasting insulin (mean ± SEM, two-way repeated measures ANOVA with Holm-Sidak post-hoc; *p<0.05, **p<0.005), and (**C**) glycosylated hemoglobin (HbA1c) at baseline (week 0; solid) and week 14 (striped) in WT and ERα KO mice provided 45% HFD (TestDiet 58V8)±17α-E2 (14.4 ppm) (mean ± SEM, two-way repeated measures ANOVA with Holm-Sidak post-hoc; **p<0.005). (**D**) Glucose tolerance testing (GTT; 1 mg/kg) (mean ± SEM, two-way repeated measures ANOVA with Holm-Sidak post-hoc; *p<0.05 between WT HFD and WT HFD+17α), and (**E**) GTT AUC during week 10 of the study (mean ± SEM, two-way ANOVA with Holm-Sidak post-hoc; *p<0.05). Age-matched, male WT, chow-fed (TestDiet 58YP) mice were also evaluated as a normal-weight reference group and the corresponding means are depicted as dashed gray lines. n = 9–10 (WT HFD), 8–10 (WT HFD+17α), 9–10 (KO HFD), 8–10 (KO HFD+17α), 12–15 (WT Chow).

The online version of this article includes the following figure supplement(s) for figure 3:

**Figure supplement 1.** 17α-E2 reverses obesity-related metabolic dysfunction in male WT, but not ERα KO, mice.

**Figure supplement 2.** 17α-E2 fails to alter metabolic parameters in WT or ERα KO female mice.

## 17α-E2 improves liver disease pathology in an ERα-dependent manner in male mice

We previously reported that 17α-E2 alters hepatic lipid deposition and DNA damage responses in male mice through unknown mechanisms (*Stout et al., 2017b*). In the current study, we sought to determine if these findings are mediated through ERα. We found that 17α-E2 significantly reduced liver mass and steatosis in male WT, but not ERα KO mice, as evidenced by reductions in oil-red-O positivity, fatty acid content, and triglyceride accumulation (*Figure 4*, *Figure 4—figure supplement 1*). These observations were accompanied by significant alterations in gene expression associated with de novo lipogenesis (fatty acid synthase [*Fasn*]) and β-oxidation (peroxisome proliferator-activated receptor alpha [*Pparα*]; sterol regulatory element binding transcription factor 1 [*Srebf1*]) (*Figure 4—figure supplement 1*). These findings are similar to previous reports showing that 17β-E2 acts through ERα to modulate the expression and activity of genes that regulate hepatic lipid metabolism (*Della Torre et al., 2016*; *Stubbins et al., 2012*; *Zhang et al., 2013*). Interestingly, despite seeing overall reductions in hepatic fatty acid content with 17α-E2 treatment in male WT mice (*Figure 4C*), we also observed elevations in specific fatty acids in these mice as compared to WT HFD controls. Notably, arachidonic acid (AA, 20:4n6) and docosahexaenoic acid (DHA, 22:6n3), both of which are precursors for eicosanoid, resolvin, and protectin production (*Szefel et al., 2015*; *Kohli and Levy, 2009*), were found to be increased by 17α-E2 treatment in male WT mice (*Figure 4—figure supplement 2*). Our findings are aligned with a previous report by Garratt et al. showing that 17α-E2 increases AA and DHA in liver (*Garratt et al., 2018*). None of the 17α-E2-mediated changes in fatty acid profiles were observed in male ERα KO mice receiving 17α-E2. In response to the elevations in AA and DHA with 17α-E2 treatment, we also assessed circulating eicosanoids. We found that 17α-E2 treatment also mildly altered several circulating eicosanoid concentrations in male WT mice (*Supplementary file 3*). Many of these have been linked to changes in inflammatory signaling (*Kiss et al., 2010*; *Gilroy et al., 2016*), although the role they are playing in 17α-E2-mediated effects of on metabolism and/or aging remain unclear.

Due to the association between obesity-related hepatic steatosis and the onset of fibrosis, we assessed collagen deposition by trichrome staining and found that 17α-E2 reduced this in male WT, but not ERα KO, mice (*Figure 5A*). We also observed significant suppression of several transcripts associated with liver fibrosis in male WT mice receiving 17α-E2, including collagen type 1 alpha 1

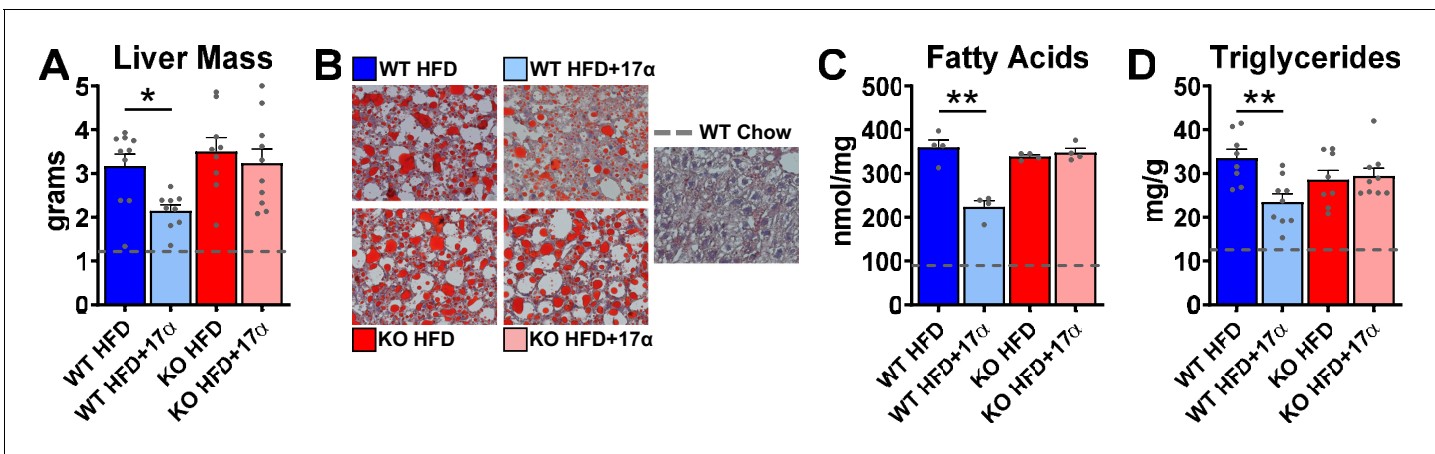

**Figure 4.** 17α-E2 reverses obesity-related hepatic steatosis in an ERα-dependent manner in male mice. (**A**) Liver mass (mean ± SEM, two-way ANOVA with Holm-Sidak post-hoc; *p<0.05), (**B**) Representative liver oil-red-O staining, (**C**) Liver fatty acids (mean ± SEM, two-way ANOVA with Holm-Sidak post-hoc; **p<0.005), and (**D**) Liver triglycerides in WT and ERα KO mice provided 45% HFD (TestDiet 58V8)±17α-E2 (14.4ppm) for 14 weeks (mean ± SEM, two-way ANOVA with Holm-Sidak post-hoc; **p<0.005). Age-matched, male WT, chow-fed (TestDiet 58YP) mice were also evaluated as a normal-weight reference group and the corresponding means are depicted as dashed gray lines. n = 4–10 (WT HFD), 4–9 (WT HFD+17α), 4–9 (KO HFD), 4–10 (KO HFD+17α), 4–15 (WT Chow).

The online version of this article includes the following figure supplement(s) for figure 4:

**Figure supplement 1.** 17α-E2 alters markers of lipid and glucose homeostasis predominantly through ERα in male mice.

**Figure supplement 2.** 17α-E2 alters the hepatic fatty acid profile in male WT, but not ERα KO, mice.

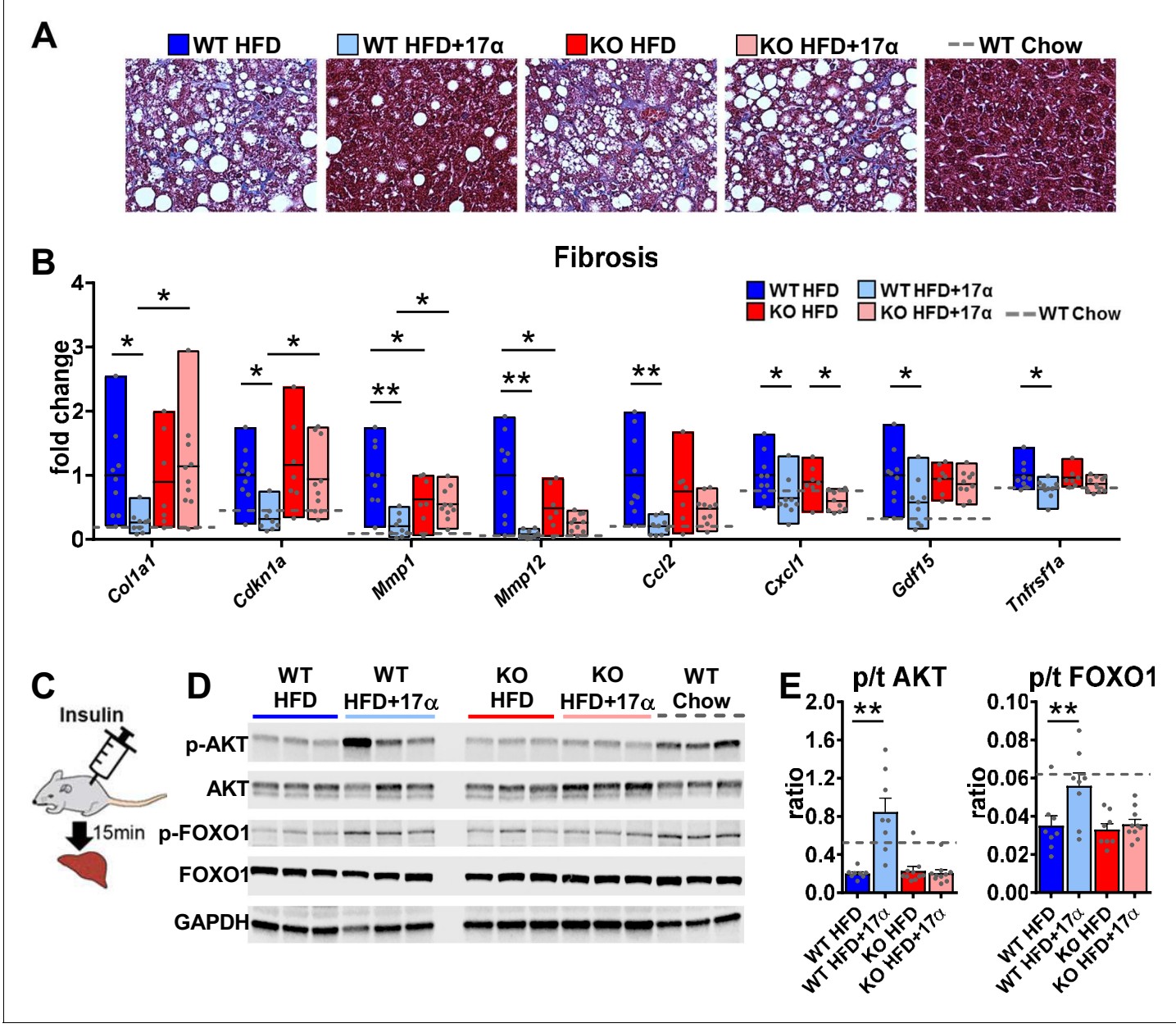

**Figure 5.** 17α-E2 reverses obesity-related liver fibrosis and insulin resistance in an ERα-dependent manner in male mice. (A) Representative liver Masson's trichrome staining for collagen and (B) Liver transcriptional markers of fibrosis in WT and ERα KO mice provided 45% HFD (TestDiet 58V8)± 17α-E2 (14.4ppm) for 14 weeks (box plots depict total range of fold changes in gene expression with mean shown as a horizontal black line, Benjamini–Hochberg multiple testing correction, two-way ANOVA with Holm-Sidak post-hoc; *p<0.05, **p<0.005). (C) Schematic of in vivo insulin stimulation (2mU/g) in fasting mice, (D) Representative liver immunoblots, and (E) Quantification of phospho/total (p/t) AKT (pS473) and FOXO1 (pS256) in WT and ERα KO mice provided 60% HFD (TestDiet 58Y1)±17α-E2 (14.4ppm) for 12 weeks (mean ± SEM, Benjamini–Hochberg multiple testing correction, two-way ANOVA with Holm-Sidak post-hoc; **p<0.005). Age-matched, male WT, chow-fed (TestDiet 58YP) mice were also evaluated as a normal-weight reference group and the corresponding means are depicted as dashed gray lines. n = 7–10 (WT HFD), 8–9 (WT HFD+17α), 7–10 (KO HFD), 10 (KO HFD +17α), 7–11 (WT Chow).

(*Col1a1*) (*Hayashi et al., 2014*; *Lua et al., 2016*), cyclin-dependent kinase inhibitor 1A (*Cdkn1a*) (*Crary and Albrecht, 1998*; *Yang et al., 2020*), matrix metallopeptidase 1 (*Mmp1*) (*Lichtinghagen et al., 2003*), matrix metallopeptidase 12 (*Mmp12*) (*Madala et al., 2010*), monocyte chemoattractant protein 1 (*Ccl2*) (*Glass et al., 2018*; *Baeck et al., 2012*), C-X-C motif chemokine ligand 1 (*Cxcl1*) (*Yang et al., 2017*), growth differentiation factor 15 (*Gdf15*) (*Koo et al., 2018*), and

TNF receptor superfamily member 1A (*Tnfrsf1a*) (*Grattagliano et al., 2019*; *Figure 5B*). Transcripts shown to be predicative of hepatic insulin resistance (follistatin [*Fst*], inhibin subunit beta E [*Inhbe*], insulin receptor substrate 2 [*Irs2*]) (*Tao et al., 2018*; *Parks et al., 2015*) and gluconeogenic plasticity (phosphoenolpyruvate carboxykinase 1 [*Pck1*], pyruvate kinase [*Pkm*]) (*Xiong et al., 2011*) were also beneficially modulated by 17α-E2 in male WT mice (*Figure 4—figure supplement 1*). To confirm that 17α-E2 improves hepatic insulin sensitivity, we also evaluated phosphorylation status of AKT and FOXO1 in livers from male WT and ERα KO mice following the administration of an insulin bolus (*Figure 5C*). We found dramatic improvements in phosphorylated AKT (pS473) and FOXO1 (pS256) in male WT mice treated with 17α-E2 (*Figure 5D–E*), whereas these benefits were not observed in male ERα KO mice. Our findings are aligned with previous reports demonstrating that hepatic ERα plays a critical role in regulating insulin sensitivity in the liver of male mice (*Yan et al., 2019*; *Guillaume et al., 2019*; *Qiu et al., 2017*; *Zhu et al., 2014*). Collectively, these findings suggest that the liver is highly responsive to 17α-E2 and that hepatic ERα is likely the signaling mechanism by which 17α-E2 prevents and/or reverses steatosis, fibrosis, and insulin resistance.

Despite our findings demonstrating that 17α-E2 reduces calorie intake and improves liver disease parameters in male mice in an ERα-dependent manner, it has historically been unclear if the benefits attributed to 17α-E2 occur primarily due to long-term reductions in calorie intake. Moreover, it remains unclear if 17α-E2 acts in a tissue-specific manner and if these observations would also occur in other mammalian species. To address these questions, we subsequently evaluated the effects of acute 17α-E2 administration during hyperinsulinemic-euglycemic clamps in male WT outbred rats. These experiments allowed us to evaluate tissue-specific insulin-sensitivity following acute 17α-E2 exposure, thereby circumventing long-term effects of the compound including reductions in calorie intake.

## Acute 17α-E2 administration improves hepatic insulin sensitivity in male rats

The hyperinsulinemic-euglycemic clamp is the gold-standard for assessing insulin action in vivo (*Ayala et al., 2010*). Animals are fasted overnight prior to receiving a constant infusion of insulin and a variable infusion of [3-$^3$H] glucose to maintain a euglycemia throughout the clamping period. Blood samples are frequently obtained to assess glucose concentration and adjust glucose infusion rates (GIRs) to maintain euglycemia, thereby allowing the calculation of insulin sensitivity to be done. Our first set of experiments in male rats sought to determine if acute peripheral infusions of 17α-E2 modulates metabolic parameters during hyperinsulinemic-euglycemic clamps (*Figure 6A*). We found that acute peripheral administration of 17α-E2 significantly increased systemic insulin responsiveness as compared to vehicle controls, which is indicated by increased GIRs (*Figure 6B*). These studies also determined that peripheral 17α-E2 administration robustly suppressed hepatic gluconeogenesis as compared to vehicle controls ($R_a$; *Figure 6C–D*), whereas glucose disposal rates ($R_d$; *Figure 6E*) were essentially identical between groups under clamped conditions. These data indicate that 17α-E2 beneficially modulates metabolic parameters independent of reductions in calorie intake and adiposity. Furthermore, these findings strongly suggest that the liver is a primary site where 17α-E2 acts to improve metabolic homeostasis due to gluconeogenesis being tightly controlled by hormonal actions on hepatocytes (*Zhang et al., 2018*). However, it also well established that the hypothalamus can directly modulate gluconeogenesis in the liver through hepatic innervation (*Timper and Brüning, 2017*); therefore, we sought to determine if acute intracerebroventricular (ICV) delivery of 17α-E2 (*Figure 6F*) could modulate metabolic parameters similarly to that observed during peripheral 17α-E2 administration. Interestingly, we found that central administration of 17α-E2 essentially phenocopied the effects of peripheral 17α-E2 infusion with regard to GIRs and suppression of hepatic gluconeogenesis (*Figure 6G–I*). These findings suggest that 17α-E2 likely acts through hypothalamic neurons to regulate hepatic gluconeogenesis. Indeed, agouti-related peptide/neuropeptide Y (AgRP/NPY) and pro-opiomelanocortin (Pomc) neurons are known to regulate hepatic glucose production (*Könner et al., 2007*; *Ruud et al., 2017*; *Pocai et al., 2005a*; *Pocai et al., 2005b*; *Dodd et al., 2018*) and both neuronal populations express ERα (*Smith et al., 2013*; *Skinner and Herbison, 1997*; *Xu et al., 2011*; *Acosta-Martinez et al., 2007*; *Stincic et al., 2018*; *Kelly and Rønnekleiv, 2015*; *Smith et al., 2014*). Collectively, the hyperinsulinemic-euglycemic clamp studies revealed that 17α-E2 definitively modulates metabolic homeostasis in an acute manner and suggests

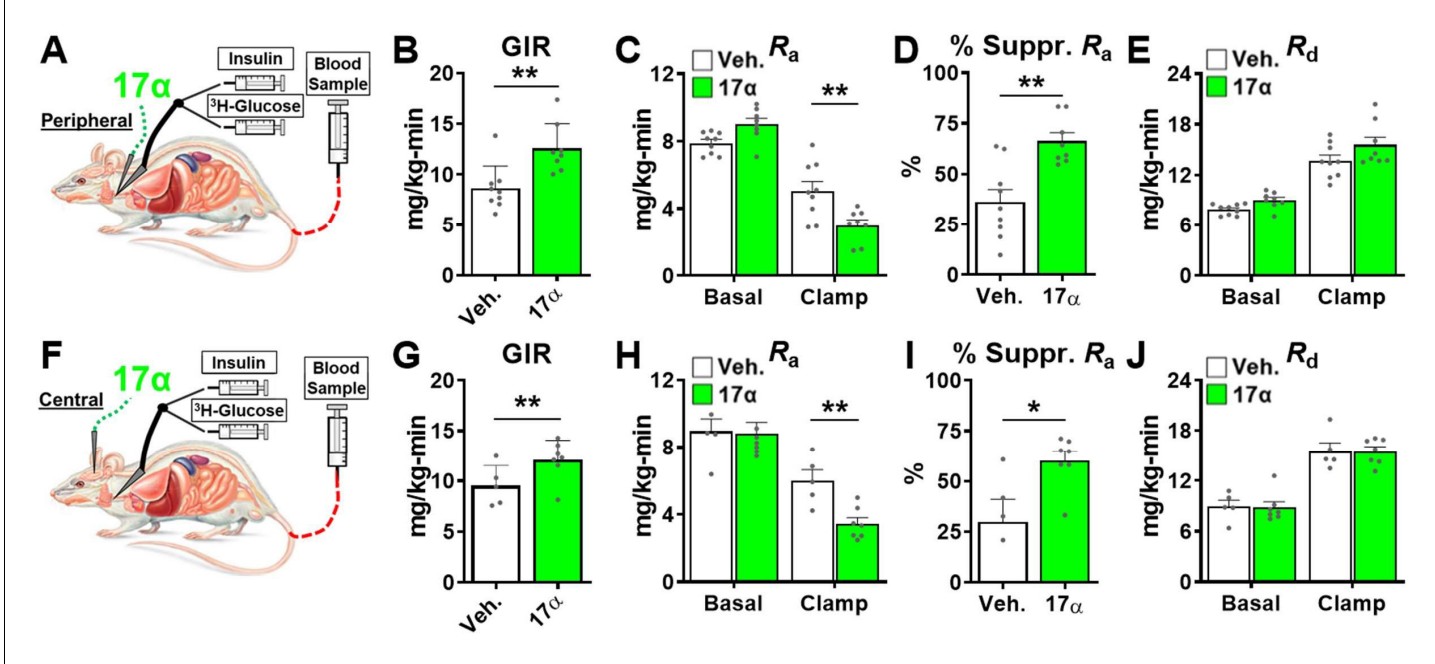

**Figure 6.** Acute delivery of 17α-E2 improves hepatic insulin sensitivity. (**A**) Schematic of peripheral 17α-E2 infusions (or vehicle) during hyperinsulinemic-euglycemic clamps, (**B**) glucose infusion rates (GIR) (mean ± SEM, unpaired Student's t-test; **p<0.005), (**C**) rate of glucose appearance ($R_a$; hepatic glucose production) (mean ± SEM, unpaired Student's t-test on Clamp; **p<0.005), (**D**) % suppression of hepatic glucose production (mean ± SEM, unpaired Student's t-test; **p<0.005), and (**E**) rate of glucose disappearance ($R_d$; peripheral glucose disposal) in 6 month old, male, FBN-F1 hybrid rats (mean ± SEM, unpaired Student's t-test on Clamp). (**F**) Schematic of ICV (central) 17α-E2 infusions (or vehicle) during hyperinsulinemic-euglycemic clamps, (**G**) GIR (mean ± SEM, unpaired Student's t-test; *p<0.05), (**H**) $R_a$ (mean ± SEM, unpaired Student's t-test on Clamp; **p<0.005), (**I**) % suppression glucose production (mean ± SEM, unpaired Student's t-test; *p<0.05), and (**J**) $R_d$ in 6-month-old, male, FBN-F1 hybrid rats (mean ± SEM, unpaired Student's t-test on Clamp). n = 5–9 (Veh.), 7–8 (17α).

that the liver and hypothalamus are two primary sites of action for the regulation of metabolic parameters by 17α-E2.

## Discussion

17α-E2 has recently been found to increase median lifespan in male mice through uncharacterized mechanisms (*Strong et al., 2016*; *Harrison et al., 2014*). We and others have shown that metabolic improvements by 17α-E2 may underlie the lifespan extending effects. In these studies, we sought to determine the role of ERα in 17α-E2-mediated transcriptional effects in vitro and metabolic effects in vivo. Although previous studies have shown that 17α-E2 has limited binding affinity for ERα, it remains unclear if 17α-E2 can induce transcriptional and physiological alterations in this manner. Given the close association between metabolic improvements and ERα activity, we hypothesized that 17α-E2 signals through ERα to elicit beneficial health outcomes. In these studies, we utilized U2OS cells stably expressing ERα and ERα global knockout mice to assess the involvement of this receptor in mediating 17α-E2 effects. Results from these studies demonstrate that ERα plays a pivotal role in 17α-E2-mediated effects on genomic activity and metabolism. Moreover, these data suggest that ERα may be a target for the treatment of aging and chronic diseases in males.

Given the similarities between the metabolic benefits observed in vivo with 17α-E2 treatment and the established body of literature linking ERα activity to systemic metabolic regulation (*Barros and Gustafsson, 2011*), we utilized a well-established cell line model to globally assess the ERα cistrome and transcriptome following 17α-E2 and 17β-E2 treatment. We found that, regardless of dose, 17α-E2 and 17β-E2 elicited the same pattern of ERα genomic binding loci and these loci shared the same DNA motif enrichments. Additionally, activation and suppression of gene expression were similar with both 17α-E2 and 17β-E2 exposure and were independent of dosage. This provides strong evidence that 17α-E2 is signaling through ERα to elicit beneficial outcomes, which is contrary to

what other reports have suggested (*Harrison et al., 2014*; *Garratt et al., 2017*; *Garratt et al., 2018*; *Toran-Allerand et al., 2002*; *Toran-Allerand et al., 2005*; *Toran-Allerand, 2005*). Toran-Allerand et al. reported that 17α-E2 signals through a novel receptor in the brain, which they termed ER-X (*Toran-Allerand et al., 2002*; *Toran-Allerand, 2005*). Although our findings appear to dispute this notion, several reports have shown that ERα exists and functions as multiple alternatively spliced variants (*Flouriot et al., 1998*; *Wang et al., 2005*; *Taylor et al., 2010*; *Zhang et al., 2016*; *Lin et al., 2013*). Therefore, we speculate that ER-X may have been a truncated, alternatively spliced, form of ERα, which nonetheless causes the same genomic and transcriptomic effects. These findings led us to investigate how ERα may modulate 17α-E2-induced benefits in vivo using ERα global KO mice.

In alignment with our previous reports (*Stout et al., 2017b*; *Steyn et al., 2018*), 17α-E2 reduced calorie intake, body mass, adiposity, and obesity-related metabolic perturbations in male WT mice. Conversely, 17α-E2 failed to elicit these beneficial effects in ERα KO mice, further supporting our hypothesis that ERα is the receptor by which 17α-E2 signals to induce beneficial metabolic outcomes. These observations are similar to how ERα is known to mediate the actions of endogenous estrogens on metabolic parameters in females (*Barros and Gustafsson, 2011*). In particular, 17β-E2 acts through ERα to regulate systemic insulin sensitivity, lipid distribution, thermogenesis, and hypothalamic anorexigenic pathways (*Barros and Gustafsson, 2011*; *Stincic et al., 2018*; *López and Tena-Sempere, 2015*). The loss of endogenous estrogen action due to menopause in humans or ovariectomy (OVX) in rodents eliminates these beneficial effects and elicits metabolic perturbations (*Stefanska et al., 2015*). Moreover, OVX following sexual maturation has also been shown to reduce lifespan in female mice (*Benedusi et al., 2015*), indicating that endogenous estrogens regulate lifespan in females; which we surmise is at least partially mediated through ERα.

In the current study, 17α-E2 failed to induce beneficial metabolic effects in female mice of either genotype, which we postulate is due to endogenous 17β-E2 saturating ERα in female WT mice, thereby limiting the potential benefits of 17α-E2 treatment. This interpretation is supported by our recent report showing that OVX renders WT female mice responsive to the beneficial effects of 17α-E2 on adiposity and bone mass (*Mann et al., 2020*), both of which are regulated by ERα activity (*Heine et al., 2000*; *Khosla and Monroe, 2018*). In males, very few studies have evaluated the role of ERα in metabolism, although a few recent reports have suggested that ERα plays tissue-specific roles, particularly in the liver, by regulating glucoregulatory pathways (*Allard et al., 2019*; *Yan et al., 2019*; *Guillaume et al., 2019*; *Qiu et al., 2017*; *Meda et al., 2020*; *Zhu et al., 2014*). These studies, coupled with our current findings, led us to speculate that 17α-E2 may be signaling through ERα in the liver to reverse metabolic disease and potentially extend healthspan and/or lifespan in males.

The liver is a major regulator of systemic metabolic homeostasis. Obesity and advancing age often promote a variety of liver conditions, including steatosis, fibrosis, and insulin resistance; all of which are associated with hallmarks of aging (*Hunt et al., 2019*), including cellular senescence (*Ogrodnik et al., 2017*), epigenetic alterations (*Horvath et al., 2014*), and dysregulated nutrient-sensing (*López-Otín et al., 2013*). We have previously shown that 17α-E2 can reduce hepatic steatosis, hepatic insulin resistance, and hepatocyte DNA damage through unknown mechanisms (*Stout et al., 2017b*). In the present study, we sought to determine if these findings are mediated through ERα. We found that 17α-E2 dramatically reduced liver mass and lipid content. As expected, these observations were not seen in ERα KO mice, providing further support for the hypothesis that 17α-E2 regulates systemic metabolic parameters through ERα. Interestingly, our findings suggest that 17α-E2 suppresses de novo lipogenesis and increases β-oxidation, predominantly in an ERα-dependent manner. This is aligned with previous reports showing that 17β-E2 can modulate hepatic lipid dynamics through both genomic and non-genomic actions (*Pedram et al., 2016*), leading to altered expression of rate limiting enzymes that control de novo lipogenesis (*Zhang et al., 2013*) and β-oxidation (*Camporez et al., 2013*). Reports have also shown that 17β-E2 can increase triglyceride export, thereby decreasing hepatic lipid deposition (*Zhu et al., 2013*). Although we did not directly assess cholesterol profiles in these studies, we speculate that 17α-E2 may partially reduce hepatic steatosis by increasing VLDL synthesis and/or triglyceride incorporation into VLDL. Additional studies will be needed to confirm how 17α-E2 alters hepatic lipoprotein dynamics.

Hepatic steatosis promotes liver fibrosis, which exacerbates hepatic insulin resistance (*Kim et al., 2015*). Endogenous estrogens and hormone replacement therapies in post-menopausal women

have been shown to serve a protective role on liver function (*Stubbins et al., 2012*; *Camporez et al., 2013*; *Zhu et al., 2013*; *Rossi et al., 2004*; *Brussaard et al., 1997*). Additionally, male humans are at a higher risk of developing hepatic steatosis and fibrosis as compared to age-matched females (*Guy and Peters, 2013*; *GBD 2017 Cirrhosis Collaborators and Collaborators, 2020*). In addition to reducing hepatic lipid deposition in male WT mice in the current studies, 17α-E2 also dramatically suppressed transcriptional and histological markers of hepatic fibrosis in an ERα-dependent manner. We also determined that 17α-E2 improved hepatic insulin sensitivity in male WT, but not ERα KO, mice. Several transcriptional markers associated with hepatic insulin resistance were suppressed in male WT mice receiving 17α-E2, whereas this suppression was almost entirely absent in ERα KO mice. Subsequent studies employing insulin stimulation prior to euthanasia also revealed that 17α-E2 robustly increased liver AKT and FOXO1 phosphorylation in male WT mice, indicating a reversal of obesity-related hepatic insulin resistance and increased control of gluconeogenesis. These findings clearly demonstrate that 17α-E2 modulates hepatic insulin sensitivity in an ERα-dependent manner. These observations are aligned with previous reports showing that 17β-E2 acts through ERα to improve glucoregulation (*Meda et al., 2020*; *Zhu et al., 2013*). This provides further support that 17α-E2 is eliciting metabolic improvements through ERα that are specific to the liver. Therefore, hepatic ERα may be a promising target for the development of therapeutics to alleviate metabolic disease in males. Future studies utilizing cell-type-specific ablation of ERα in the liver will be needed to unravel these possibilities.

Despite the robust effects of 17α-E2 on liver function, it remained unclear if 17α-E2 directly modulates hepatic insulin sensitivity or if these benefits were a secondary response to prolonged reductions in calorie intake, adiposity, and lipid redistribution. To test this, we performed hyperinsulinemic-euglycemic clamps, in conjunction with acute infusions of 17α-E2 in male WT rats. We found that peripheral infusions of 17α-E2 improved hepatic insulin sensitivity almost immediately, as evidenced by a greater suppression of hepatic glucose production in rats receiving 17α-E2 as compared to vehicle controls. Additionally, we did not observe improvements in glucose disposal, thereby indicating that 17α-E2 does not acutely increase systemic insulin-stimulated glucose uptake. This observation is aligned with recent literature demonstrating limited involvement of ERα in skeletal muscle insulin sensitivity (*Iñigo et al., 2020*). These data indicate that 17α-E2 primarily alters systemic metabolic homeostasis through the modulation of hepatic gluconeogenesis, which is known to account for 76–87% of glucose production in the body (*Cherrington et al., 1994*). Although these studies are suggestive of direct actions in the liver, 17α-E2 also has the ability to cross the blood brain barrier and elicit responses in the hypothalamus (*Steyn et al., 2018*). Given that the hypothalamus can regulate hepatic glucose production (*Könner et al., 2007*; *Ruud et al., 2017*; *Pocai et al., 2005a*; *Pocai et al., 2005b*; *Dodd et al., 2018*; *Timper and Brüning, 2017*), we also evaluated systemic insulin sensitivity following central administration of 17α-E2. These experiments essentially phenocopied the results of the peripheral 17α-E2 infusions, suggesting that the suppression of hepatic gluconeogenesis by 17α-E2 is at least partially mediated by the hypothalamus. A multitude of studies have shown that the arcuate nucleus (ARC) of the hypothalamus plays a critical role in the regulation of hepatic gluconeogenesis through autonomic regulation and vagus nerve activity (*Ruud et al., 2017*; *Pocai et al., 2005a*; *Zhang et al., 2018*; *Brandt et al., 2018*). Multiple neuronal populations within the ARC are known to be involved in the regulation of hepatic glucose production, including Pomc (*Dodd et al., 2018*) and AgRP/NPY (*Könner et al., 2007*; *Ruud et al., 2017*; *Pocai et al., 2005a*; *Pocai et al., 2005b*). Similarly, we have previously shown that the effects of 17α-E2 on calorie intake and adiposity are dependent upon functional Pomc neurons, thereby providing evidence that 17α-E2 can act through the hypothalamus to mediate systemic metabolic parameters (*Steyn et al., 2018*). However, in the absence of functional Pomc neurons, 17α-E2 was still able to reduce fasting glucose and insulin, suggesting that 17α-E2 modulates peripheral metabolism through multiple mechanisms, which may include alternative neuronal populations. Given that both Pomc (*Xu et al., 2011*) and AgRP/NPY (*Smith et al., 2013*; *Skinner and Herbison, 1997*; *Acosta-Martinez et al., 2007*; *Stincic et al., 2018*; *Kelly and Rønnekleiv, 2015*; *Smith et al., 2014*; *Sar et al., 1990*) neurons express ERα, regulate systemic metabolic parameters, and modulate feeding circuitry in a coordinated counter-regulatory fashion, it remains unclear whether 17α-E2 is altering hepatic and systemic metabolic parameters through Pomc and/or AgRP/NPY neurons. Interestingly, a recent report from Debarba et al. demonstrated that 17α-E2 increased hypothalamic ERα expression in the ARC (*Debarba, 2020*), which further suggests that 17α-E2 signals through

ERα in the hypothalamus. Future studies utilizing hypothalamic cell-type-specific ERα KO models will be needed to disentangle which populations of neurons are required for 17α-E2 to control food intake, peripheral glucose homeostasis, and insulin sensitivity.

Collectively, our findings strongly suggest that 17α-E2 acts through ERα in the liver and/or hypothalamus to modulate metabolic parameters. However, our findings are in contrast to other reports suggesting that 17α-E2 elicits health benefits by modulating androgen metabolism (*Garratt et al., 2017*; *Garratt et al., 2018*; *Garratt and Stout, 2018*). Garratt et al. reported that responsiveness to 17α-E2 was significantly attenuated in castrated male mice (*Garratt et al., 2018*), which the authors proposed may indicate 17α-E2 acts as a 5α-reductase inhibitor (*Schriefers et al., 1991*) to prevent the conversion of testosterone into dihydrotestosterone (DHT). 17α-E2 is known to be a mild 5α-reductase inhibitor that is prescribed as a topical treatment for androgenetic alopecia (*Orfanos and Vogels, 1980*). 5α-Reductase inhibition could conceivably elicit beneficial metabolic effects by either reducing the concentration of DHT, which has been shown to decrease adiposity (*Movérare-Skrtic et al., 2006*; *Bolduc et al., 2004*), or by promoting greater aromatization of testosterone to 17β-E2 (*Veldhuis et al., 2009*), which has been linked to improvements in metabolic parameters (*Rubinow, 2017*). If true, this would imply that the benefits of 17α-E2 are occurring in an indirect manner. However, the dose of 17α-E2 used in the vast majority of these studies, does not induce dramatic feminization of the sex hormone profiles in male mice (*Stout et al., 2017b*), which leads us to speculate that 17a-E2 is acting in a direct manner through ERα rather than indirectly through androgen modulation. Furthermore, studies in male rodents (*Livingstone et al., 2015*; *Dowman et al., 2013*) and humans *Wei et al., 2019* demonstrate that 5α-reductase inhibition or deficiency increases insulin resistance and hepatic steatosis and fibrosis, which are contradictory to the effects of 17α-E2 treatment in all of our studies utilizing male mice (*Stout et al., 2017b*; *Steyn et al., 2018*; *Miller, 2020*; *Sidhom et al., 2020*). Despite these contrasting observations, the studies by Garratt et al. do provide important insights into the interconnected and underappreciated relationship between androgen- and estrogen-signaling pathways and their roles in metabolism and aging. For instance, several recent reports have demonstrated interactions between the androgen receptor (AR) and ERα (*D'Amato et al., 2016*; *Panet-Raymond et al., 2000*; *Peters et al., 2009*), which suggests that modulation of one may affect function of the other. Additional factors to consider when comparing and contrasting our studies from those of Garratt et al. are differences in the length of study, age, and obesity status of the mice, and counterregulatory and/or compensatory effects of castration. Notably, it is plausible that 17α-E2 could be inducing metabolic benefits and lifespan-extending effects through several distinct mechanisms, including direct actions through ERα, suppression of DHT production, and/or aromatization of testosterone. Future studies will be needed to discern the potentially interdependent nature of 17α-E2 actions on ERα and androgen metabolism in metabolic improvement and lifespan extension.

There are a few notable caveats to our studies. First, we utilized constitutive global ERα KO mice, which have been shown to display varying degrees of compensatory ERβ activity due to the absence of ERα during development (*Sánchez-Criado et al., 2012*; *Rosenfeld et al., 1998*). However, if compensatory ERβ expression was playing a role in our study, we likely would not see a complete attenuation of 17α-E2-mediated effects. As such, the results of our studies clearly indicate that ERα is the primary receptor by which 17α-E2 signals. Another potential concern of the model is that ERα KO mice are known to have elevated endogenous testosterone levels (*Gould et al., 2007*), although the studies by Garratt et al. would suggest that higher testosterone levels could potentially render the male mice more responsive to 17α-E2, whereas we observed the opposite. Future studies utilizing inducible Cre models to knockdown ERα post-sexual development may be considered if it is determined that Cre induction and subsequent ERα ablation is consistent throughout multiple organ systems, which has been shown to be inconsistent in other reports (*Murray et al., 2012*). Despite these minor concerns related to the model, the use of the constitutive global ERα KO was undoubtedly the best option for these studies. However, it must also be noted that female mice present a greater phenotypic response than males to the ablation of ERα, thereby exacerbating obesity and metabolic dysfunction which makes comparisons to female WT mice as well as their male littermates problematic (*Manrique et al., 2012*; *Vidal et al., 1999*). For this reason, we chose not to provide HFD to female mice in these studies. Regardless, 17α-E2 still failed to elicit beneficial responses in female mice of either genotype (WT or ERα KO), which is aligned with previous reports demonstrating a lack of effect of 17α-E2 in intact females (*Garratt et al., 2017*; *Garratt et al., 2018*). We also

recently reported that OVX renders female mice responsive to several of the benefits conferred by 17α-E2 treatment in male mice (*Mann et al., 2020*). These observations, coupled with the genomic data presented herein, support the hypothesis that endogenous 17β-E2 actions on ERα diminishes potential benefits of 17α-E2 in intact female mice. However, future studies will be needed to definitively determine if female mice subjected to diet-induced obesity will display responsiveness to 17α-E2 once severe metabolic dysfunction has emerged. Lastly, the current studies were relatively short in duration and it remains unclear if metabolic improvements with 17α-E2 treatment are required for the lifespan extension effects of the compound. Although several other studies have evaluated the long-term effects of 17α-E2 (*Strong et al., 2016*; *Harrison et al., 2014*; *Garratt et al., 2017*; *Garratt et al., 2018*), a shorter treatment duration was effective for testing our hypothesis in these studies. Similarly, given the close relationship between metabolic homeostasis, sex hormones, and longevity (*López-Otín et al., 2013*; *Barros and Gustafsson, 2011*), we surmise that future studies evaluating the effects of ERα on male lifespan in the presence or absence of 17α-E2 will be needed. Although our current report does not provide direct evidence that ERα modulates the lifespan extending effects of 17α-E2, it does provide insight into the involvement of hepatic and/or hypothalamic ERα on 17α-E2-mediated metabolic effects in male mice.

In summary, the data presented herein are the first to show that 17α-E2 and 17β-E2 induce nearly identical ERα chromatin association patterns and transcriptional activity. Moreover, we demonstrate that the metabolic benefits of 17α-E2 in male mice are ERα-dependent. We also provide evidence that strongly suggests 17α-E2 acts through the liver and hypothalamus to regulate metabolic homeostasis in male mice. These effects were mirrored by studies in male WT rats receiving 17α-E2, indicating that 17α-E2 can modulate metabolism almost instantaneously and that these effects are not limited to a single mammalian species. Future studies will be needed to confirm that 17α-E2 acts predominantly through ERα in a cell-type-specific manner in the liver and hypothalamus to modulate systemic metabolic homeostasis. It is also imperative that we determine if ERα exclusively modulates the lifespan-extending effects of 17α-E2 in male mice. Another potential avenue of investigation that remains unresolved is whether 17α-E2 acts through ERα in a genomic or non-genomic manner to modulate health parameters. Potential interactions between androgen and estrogen signaling must also be considered when evaluating the effects of 17α-E2 on metabolism and lifespan. These studies will provide additional insight into mechanisms of metabolic improvement and lifespan extension by 17α-E2. Our studies provide critical insight into the molecular mechanisms by which 17α-E2 elicits metabolic benefits in males, which were previously unknown and may underlie its lifespan-extending effects.

# Materials and methods

**Key resources table**

| Reagent type (species) or resource | Designation | Source or reference | Identifiers | Additional information |
|---|---|---|---|---|
| Genetic reagent (*M. musculus*) | B6N(Cg)-Esr1tm4.2Ksk/J | The Jackson Laboratory | Stock No:026176; RRID:IMSR_JAX:026176 | ERα (*Esr1*) KO mice |
| Cell line (*Homo sapien*) | U2OS Cells | ATCC | HTB-96; RRID:CVCL_0042 | PMID:15802376 PMID:14505348 |
| Antibody | anti-FLAG M2 (Mouse monoclonal) | Sigma-Aldrich | F1804 | IP: 1 uL per pull-down (1 mg/mL) |
| Commercial assay or kit | Protein G Dynabeads | Applied Biosystems/ Thermofisher Scientific | 10003D | IP: 30 uL per IP |
| Chemical compound, drug | 17α-E2 | Steraloids, Inc | E0870-000 | |
| Chemical compound, drug | Novolin R 100 U/ml | Novolin | | 2mU/g |
| Other (diet) | Chow; TestDiet 58YP | TestDiet | TestDiet 58YP | |
| Other (diet) | HFD; TestDiet 58V8 | TestDiet | TestDiet 58V8 | HFD 45% by kcal |
| Other (diet) | HFD; TestDiet 58Y1 | TestDiet | TestDiet 58Y1 | HFD 60% by kcal |

*Continued on next page*

*Continued*

| Reagent type (species) or resource | Designation | Source or reference | Identifiers | Additional information |
|---|---|---|---|---|
| Commercial assay or kit | Mouse Ultrasensitive Insulin ELISA | ALPCO | Cat# 80-INSMSU-E01; RRID:AB_2792981 | |
| Commercial assay or kit | Free Glycerol Agent | Sigma-Aldrich | Sigma F6428 | |
| Commercial assay or kit | Triglyceride Reagent | Sigma-Aldrich | Sigma F6428 | |
| Commercial assay or kit | Glycerol Standard | Sigma-Aldrich | Sigma G1394 | |
| Antibody | anti-pS473 AKT (Rabbit polyclonal) | Abcam | Cat# ab81283; RRID:AB_2224551 | WB: (1:3000) |
| Antibody | Anti-pan-AKT (Rabbit polyclonal) | Abcam | Cat# ab179463; RRID:AB_2810977 | WB (1:10000) |
| Antibody | Anti-pS256 FOX01 (Rabbit polyclonal) | Abcam | Cat# ab131339; RRID:AB_11159015 | WB (1:1000) |
| Antibody | Anti-FOX01a (Rabbit polyclonal) | Abcam | Cat# ab52857; RRID:AB_869817 | WB (1:1000) |
| Antibody | Anti-GAPDH (Rabbit polyclonal) | Abcam | Cat# ab9485; RRID:AB_307275 | WB (1:2500) |
| Antibody | Anti-Rabbit IgG, IRDye 800 CW | LI-COR | Cat# 926–32211; RRID:AB_621843 | WB (1:15000) |
| Commercial assay or kit | TaqMan Gene Expression Master Mix | Applied Biosystems/ Thermofisher Scientific | 4369542 | |
| Sequenced-based reagent | qPCR primer Mmp1 | Integrated DNA Technologies | Mm.PT.58.42286812 Ref Seq: NM_008607(1) | Exon 5–6 |
| Sequenced-based reagent | qPCR primer Mmp12 | Integrated DNA Technologies | Mm.PT.58.31615472 Ref Seq: NM_008605(1) | Exon 8–9 |
| Sequenced-based reagent | qPCR primer Ccl2 | Integrated DNA Technologies | Mm.PT.58.42151692 Ref Seq: NM_011333(1) | Exon 1–3 |
| Sequenced-based reagent | qPCR primer Srebf1 | Integrated DNA Technologies | Mm.PT.58.8508227 Ref Seq: NM_011480(1) | Exon 1–2 |
| Sequenced-based reagent | qPCR primer Pck1 | Integrated DNA Technologies | Mm.PT.58.11992693 Ref Seq: NM_011044(1) | Exon 3–4 |
| Sequenced-based reagent | qPCR primer Cdkn1a | Integrated DNA Technologies | Mm.PT.58.17125846 Ref Seq: NM_007669(1) | Exon 2–3 |
| Sequenced-based reagent | qPCR primer Pparα | Integrated DNA Technologies | Mm.PT.58.9374886 Ref Seq: NM_001113418(2) | Exon 8–9 |
| Sequenced-based reagent | qPCR primer Cxcl1 | Integrated DNA Technologies | Mm.PT.58.42076891 Ref Seq: NM_008176(1) | Exon 2–4 |
| Sequenced-based reagent | qPCR primer Col1a1 | Integrated DNA Technologies | Mm.PT.58.7562513 Ref Seq: M_007742(1) | Exon 1–2 |
| Sequenced-based reagent | qPCR primer Tnfrsf1a | Integrated DNA Technologies | Mm.PT.58.28810479 Ref Seq: NM_011609(1) | Exon 5–7 |
| Software, algorithm | SigmaPlot 12.5 | Systat Software | RRID:SCR_003210 | statistical analyses |
| Software, algorithm | ImageJ | ImageJ | RRID:SCR_003070 | histological quantification |
| Software, algorithm | Image Studio | LI-COR | RRID:SCR_015795 | western blot quantification |
| Software, algorithm | RStudio | GenomicAlignments DiffBind DESeq2 GenomicRanges | RRID:SCR_000432 | Peak Calling Differential expression Differential binding |
| Software, algorithm | Bowtie2 MACS2 Bedtools Samtools Picard-tools Trimmomatic | Bowtie2 MACS2 Bedtools Samtools Picard-tools Trimmomatic | | Alignment, Peak Calling, trimming, duplicate identification |

## U2OS cells

U2OS osteosarcoma cells stably expressing flag-tagged ERα (U2OS-ERα) under the control of doxy-cycline (dox)-inducible promoter (*Monroe et al., 2003*) were utilized for the studies described here. U2OS cells were originally purchased from ATTC and were authenticated using IDEXX BioAnalytics (Westbrook, ME). Cells were also regularly checked for mycoplasma contamination using a PCR-based mycoplasma detection kit from SouthernBiotech (Birmingham, AL) and were confirmed to be negative. U2OS-ERα cells were cultured in phenol-free αMEM medium supplemented with 10% HyCloneTM charcoal/dextran stripped FBS (GE Healthcare Life Sciences, Pittsburgh, PA), 1% antibi-otic/antimycotic, 5 mg/L blasticidin S, and 500 mg/L zeocin in a humidified 37°C incubator with 5% $CO_2$. Cells were plated in 12-well plates in the presence of doxycycline to induce ERα expression. The following day, cells were treated for 24 hr with 17β-E2 (10 nM) or 17α-E2 (10 nM and 100 nM) (Steraloids, Newport, RI) in charcoal-stripped FBS-containing media.

## ChIP-sequencing

To evaluate patterns of ERα binding agonized by 17α-E2 vs 17β-E2, we performed ChIP-Sequencing. U2OS-ERα cells were harvested 24 hr post-treatment and chromatin immunoprecipitation was per-formed as previously described (*Reese et al., 2018*; *Nelson et al., 2006*). Briefly, ERα was immuno-precipitated overnight at 4°C using 10 μg of Flag antibody (clone M2, Sigma-Aldrich, St. Louis, MO). Complexes bound to the antibody were captured with protein G Dynabeads (Thermo Fisher Scien-tific, Waltham, MA), extensively washed, and reverse cross-linked at 65°C overnight. DNA isolation was performed by phenol/chloroform extraction and was used for ChIP-sequencing library prepara-tion. Libraries were sequenced using paired-end 100 bp reads on the Illumina HiSeq 4000 (GSE151039). Reads were aligned to the human genome (hg19, https://genome.ucsc.edu/cgi-bin/hgGateway) using bowtie2 (*Langmead and Salzberg, 2012*) and duplicated reads were flagged with Picard-tools (http://broadinstitute.github.io/picard/). ERα-binding peaks were called using MACS2 (*Zhang et al., 2008*) with recommended settings. Peak genomic location, breadth of cover-age, and peak summit location were determined using MACS2. NarrowPeak files containing peak information were used to determine differential ERα binding. First, peaks were centralized around the summit and 250 bp flanking regions were added to the summit location to generate equal 500 bp regions across all experimental groups. Peak files were then used to extract read counts from the aligned de-duplicated BAM file using samtools (*Li et al., 2009*), read counts were then normalized to total library sequencing depth. To determine differential binding, the R package diffbind was uti-lized (*Ross-Innes et al., 2012*). Normalized read counts were log2 transformed and normalized across all experimental groups. Differential binding between treatment groups was determined using negative binomial regression models utilized in the R package DESeq2, statistical significance for pairwise comparisons between experimental groups was determined using Wald test. To account for multiple comparisons, we used Benjamini-Hochberg multiple testing correction (False-discovery rate, FDR). Motif analysis was performed using HOMER with standard settings to identified motifs. Peak regions called for each treatment group were analyzed to identify enriched motifs relative to the entire genome. For pairwise differential motif enrichment or depletion between experimental groups, we utilized a hypergeometric test with the number of sequences with a motif from each group and total number of peaks as total sample size. Motifs that appeared in less than five sequen-ces between both pairwise test groups were removed. Benjamini-Hochberg multiple testing correc-tion was utilized to control for multiple testing (FDR < 0.05).

## RNA-sequencing

U2OS-ERα cells were harvested 24 hr post-treatment and RNA was extracted using Trizol and DNase cleanup. RNA libraries were prepared with Illumina's TrueSeq RNA-seq library prep accord-ing to manufacturer protocol. Libraries were sequenced with 150 bp paired-end reads on the Illu-mina 4000 platform (Illumina, San Diego, CA) (GSE151039). Sequence quality control was performed with fastQC, Paired reads were trimmed using trimmomatic, and were aligned to the hg19 genome using STAR (*Dobin et al., 2013*). Differential expression was determined using previously described methods (*Hadad et al., 2019*). In brief, gene counts were determined with the R package Genomi-cAlignments 'summarizeOverlap' function. Gene counts were then transformed using regularized log

transformation and normalized relative to library size using the DESeq2 (*Love et al., 2014*) R package. Differential expression was determined using negative binomial generalized linear model using *counts ~ treatment* model. We performed pairwise differential expression between all experimental groups using Wald test. All comparisons were corrected for multiple testing using Benjamini-Hochberg multiple testing correction method. Differential expression significance threshold was set to FDR corrected $p < 0.05$.

## Animal study 1

To determine if ERα is the primary receptor by which 17α-E2 signals to elicit metabolic benefits in vivo, we utilized male global ERα KO and WT littermate mice. Mice were acquired from Dr. Kenneth Korach (National Institute of Environmental Health Sciences [NIEHS]) and were also bred at OUHSC by pairing ERα heterozygous KO mice (JAX; strain #026176). Mice were fed a 45% high-fat diet (HFD) (TestDiet 58V8, 35.5% CHO, 18.3% PRO, 45.7% FAT) from TestDiet (Richmond, IN) for 4 months prior to study initiation to induce obesity and metabolic perturbations. Additionally, age-matched, male WT, chow-fed mice were maintained on TestDiet 58YP (66.6% CHO, 20.4% PRO, 13.0% FAT) throughout the entire study as a healthy-weight reference group. Mice were individually housed with ISO cotton pad bedding, cardboard enrichment tubes, and nestlets at 22 ± 0.5°C on a 12:12 hr light-dark cycle. Unless otherwise noted, all mice had ad libitum access to food and water throughout the experimental timeframe. At the conclusion of the fattening period, all mice (age: 6–8 months) receiving HFD were randomized within genotype by age, body mass, fat mass, calorie intake, fasting glucose, fasting insulin, and glycosylated hemoglobin (HbA1C) into HFD or HFD+17α (TestDiet 58V8 + 17α-E2, 14.4ppm; Steraloids, Newport, RI) treatment groups for a 14-week intervention. Body mass and calorie intake were assessed daily for the first 4 weeks, followed by body mass and body composition (EchoMRI, Houston, TX) on a weekly basis. At 10 weeks post-treatment, mice were fasted for 5–6 hr and fasting glucose, fasting insulin, HbA1C, and glucose tolerance were assessed. At the conclusion of the study (14 weeks post treatment), mice were euthanized with isoflurane in the fasted state (5–6 hr). Blood was collected into EDTA-lined tubes by cardiac puncture, and plasma was collected and frozen. Tissues were excised, weighed, flash frozen, and stored at −80°C unless otherwise noted. Small sections of liver were fixed in 4% paraformaldehyde in preparation for paraffin- or cryo-embedding for future analyses. All animal procedures were reviewed and approved by the Institutional Animal Care and Use Committee at OUHSC.

## Animal study 2

Although previous studies [*Strong et al., 2016*; *Harrison et al., 2014*; *Garratt et al., 2017*; *Garratt et al., 2018*] have demonstrated minimal effects of 17α-E2 in female mice, we thought it prudent to determine if the ablation of ERα would alter female responsiveness to 17α-E2. Female WT and ERα KO mice were acquired from Dr. Kenneth Korach (National Institute of Environmental Health Sciences [NIEHS]). Female mice were maintained on Chow TestDiet 58YP (66.6% CHO, 20.4% PRO, 13.0% FAT) and were not subject to HFD feeding due to ERα KO female mice naturally displaying an obesity phenotype. Mice were individually housed with ISO cotton pad bedding, cardboard enrichment tubes, and nestlets at 22 ± 0.5°C on a 12:12 hr light-dark cycle. Unless otherwise noted, all mice had ad libitum access to food and water throughout the experimental timeframe. At age 9–11 months, female mice were randomized within genotype by age, body mass, fat mass, calorie intake, fasting glucose, fasting insulin, and glycosylated hemoglobin (HbA1C) into Chow or Chow +17α-E2 (TestDiet 58YP + 17α-E2,14.4ppm; Steraloids, Newport, RI) treatment groups. The study was terminated following a 4-week intervention due to a lack of responsiveness to 17α-E2. At the conclusion of the study, mice were euthanized with isoflurane in the fasted state (5–6 hr). Blood was collected into EDTA-lined tubes by cardiac puncture, and plasma was collected and frozen. Tissues were excised, weighed, flash frozen, and stored at −80°C. All animal procedures were reviewed and approved by the Institutional Animal Care and Use Committee at OUHSC.

## Animal study 3

To assess insulin sensitivity within the liver, an additional cohort of ERα KO and WT littermate mice were bred from mice acquired from Jackson Laboratory (JAX; strain #026176), which were generated from identical founder strains in the laboratory of Dr. Korach at NIEHS. Male ERα KO and WT

mice were fed a 60% high-fat diet (HFD; TestDiet 58Y1, 20.3% CHO, 18.1% PRO, 61.6% FAT) for 4 months prior to study initiation to induce obesity and metabolic perturbations. Additionally, as was done in Animal Study 1, age-matched, male WT, chow-fed mice were maintained on TestDiet 58YP (66.6% CHO, 20.4% PRO, 13.0% FAT) throughout the entire study as a healthy-weight reference group. Mice were group housed with corncob bedding, cardboard enrichment tubes, and nestlets at $22 \pm 0.5°C$ on a 12:12 hr light-dark cycle. Unless otherwise noted, all mice had ad libitum access to food and water throughout the experimental timeframe. At the conclusion of the fatting period, all mice (age: 6 months) receiving HFD were randomized within genotype by body mass, fat mass, calorie intake, fasting glucose, and fasting insulin into HFD or HFD+17α (TestDiet 58Y1 + 17α-E2, 14.4ppm; Steraloids, Newport, RI) treatment groups for a 12-week intervention. Prior to being euthanized, mice were fasted (5–6 hr) and IP injected with insulin (Novolin R 100 U/ml; 2mU/g) to assess insulin activity and sensitivity in tissue as previously described (*Lu et al., 2012*). Each mouse was euthanized with isoflurane 15 min following their insulin injection. Blood was collected into EDTA-lined tubes by cardiac puncture, and plasma was collected and frozen. Tissues were excised, weighed, flash frozen, and stored at −80°C unless otherwise noted. All animal procedures were reviewed and approved by the Institutional Animal Care and Use Committee at OUHSC.

## Animal study 4

Hyperinsulinemic-euglycemic clamp experiments, the gold-standard for assessing insulin sensitivity, were performed in male rats to determine if 17α-E2 can acutely modulate insulin sensitivity and glucose homeostasis. FBN-F1 hybrid male rats were acclimated to the animal facilities within the Einstein Nathan Shock Center for 2 weeks prior to undergoing surgeries in preparation for hyperinsulinemic-euglycemic clamp studies. Rats were fed Purina 5001 (58.0% CHO, 28.5% PRO, 13.5% FAT) and were individually housed with corncob bedding at $22 \pm 0.5°C$ on a 14:10 hr light-dark cycle with ad libitum access to food and water. All surgeries were conducted under 2% isoflurane. For clamp studies incorporating central infusions, rats underwent two surgical procedures. First, stereotactic placement of a steel-guide cannula (Plastics One, Roanoke, VA) reaching the 3rd ventricle was performed. The implant was secured in place with dental cement and animals were treated with analgesic as needed. Approximately 14 days later, animals underwent a second surgical procedure to place indwelling catheters into the right internal jugular vein and the left carotid artery, which was also performed for animals undergoing only peripheral clamp studies. Hyperinsulinemic-euglycemic clamp studies incorporating peripheral 17α-E2 infusions were performed as previously described (*Einstein et al., 2010*). For studies employing peripheral infusions of 17α-E2, 17α-E2 was diluted in sterile saline to a final concentration of 30 ng/μl. Beginning at t = 0 min animals received a primed-continuous infusion of saline or 30 ng/μl 17α-E2 provided as a 3 μg bolus at a rate of 20 ul/min over 5 min, followed by a continuous infusion at a rate of 0.06 ml/hr over 235 min (9.4 ng/hr) for a maintenance dose of 7 μg (total dose 10 μg). Hyperinsulinemic-euglycemic clamp studies with intracerebroventricular (ICV) infusions were performed as previously described (*Huffman et al., 2016a*). 17α-E2 powder (Steraloids, Newport, RI) was dissolved in DMSO at a concentration of 10 mg/ml and stored at −80°C. For ICV infusions, 17α-E2 was diluted in artificial cerebral spinal fluid (ACSF) to a final concentration of 2 ng/μl. Beginning at t = 0 min, animals received a primed-continuous ICV infusion of ACSF (Veh.) or 17α-E2 (17α) provided as a 15 ng bolus at a rate of 1 μl/min over 7.5 min, followed by a continuous infusion of 56.5 ng at a rate of 0.08 μl/hr over 6 hr (9.4 ng/hr) and a total dose of 71.5 ng. All animal procedures were reviewed and approved by the Institutional Animal Care and Use Committee at the Einstein College of Medicine.

## In vivo metabolic analyses in mice

To evaluate the effects of 17α-E2 on metabolic parameters in vivo, we performed several assessments of glucose homeostasis. Unless otherwise noted, all experiments requiring fasting conditions were performed in the afternoon, 5–6 hr following the removal of food at the beginning of the light-cycle as outlined elsewhere (*Ayala et al., 2010*). To ensure fasting conditions, mice were transferred to clean cages containing ISO cotton padding and clean cardboard enrichment tubes. Non-terminal blood was collected via tail snip. Fasting glucose was evaluated using a Bayer Breeze 2 Blood Glucose Monitoring System (Bayer Global, Leverkusen, Germany). Fasting insulin was evaluated using a Mouse Ultrasensitive Insulin ELISA from Alpco (Salem, NH). HbA1c was assessed by A1C-Now

Monitoring kits (Bayer, Whippany, NJ). Glucose tolerance tests were performed following a 5 hr fast using an intraperitoneal filtered dextrose injection of 1 g/kg body mass (*Huffman et al., 2016b*). Blood glucose was measured immediately pre-injection (time 0) and at 15, 30, 60, 90, and 120 min post-injection.

## Liver histology

To evaluate the effects of 17α-E2 treatment on lipid accumulation and fibrosis, we evaluated fixed liver tissue. Tissues were fixed in 4% PFA for 24 hr, cryo-embedding samples were transferred to 30% sucrose for 72 hr and embedded in OCT, paraffin-embedding samples were transferred to 1X PBS for 48 hr, then to 70% ethanol until embedding. Liver oil-red-O and Masson's trichrome staining were performed by the Oklahoma Medical Research Foundation Imaging Core Facility using previously reported methodology (*Leonard et al., 2018*; *Mehlem et al., 2013*). Oil-red-O (ORO, Sigma-Aldrich, St. Louis, MO) and H and E counterstaining were performed on cryo-embedded tissues, and were imaged within 6 hr of staining. Red lipid stain was blindly quantified from 10 images per animal using ImageJ software and presented as a lipid to total tissue ratio. Masson's trichrome staining was performed on paraffin embedded liver tissue and was used for qualitative purposes. In brief, slides were stained with Weigert's Iron Hematoxylin (Sigma-Aldrich, St. Louis, MO), washed, stained with Biebrich Scarlet-Acid Fusion (Sigma-Aldrich, St. Louis, MO), washed, stained with Phosphomolybdic Acid-Phosphotunsctic Acid, and then stained with Aniline Blue (Sigma-Aldrich, St. Louis, MO).

## Liver triglycerides

We evaluated the effects of 17α-E2 treatment on triglyceride accumulation in the liver. Liver samples (~100 mg) were homogenized on ice for 60 s in 10X (v/w) Cell Signaling Lysis Buffer (Cell Signaling, Danvers, MA) with protease and phosphatase inhibitors (Boston BioProducts, Boston, MA). Total lipid was extracted from the homogenate using the Folch method with a 2:1 chloroform-methanol mixture (*Folch et al., 1957*). Lipid was dried down using a nitrogen drier at room temperature, and resuspended in 100 µl of 3:1:1 tert-butyl alcohol-methanol-Triton X-100 solution. Final triglyceride concentrations were determined using a spectrophotometric assay with a 4:1 Free Glycerol Agent/Triglyceride Agent solution (Sigma Triglyceride and Free-Glycerol reagents, St. Louis, MO) as previously described (*Stout et al., 2011*).

## Liver fatty acids

We evaluated the effects of 17α-E2 on hepatic fatty acid content. Liver samples (~50 mg) were homogenized and on ice for 60 s in 10X (v/w) Cell Signaling Lysis Buffer (Cell Signaling, Danvers, MA) with protease and phosphatase inhibitors (Boston BioProducts, Boston, MA). Total lipid was extracted using a modified Bligh and Dyer method (*Bligh and Dyer, 1959*) (Sigma-Aldrich, St. Louis, MO). Of 15:0 and 17:0 internal standards, 50 nmol were added and acid hydrolysis/methanolysis was done to generate fatty acid methyl esters (FAMEs) (*Agbaga et al., 2018*). FAMEs were identified as previously described by GC-MS (*Agbaga et al., 2018*). A 6890N gas chromatograph with flame ionization detector (GC-FID) (Agilent Technologies) was used to quantify FAMEs (*Yu et al., 2012*). Standards 15:0 and 17:0 were used to compare and determine sample concentrations. Data is represented as the relative mole percent of each fatty acid.

## Plasma eicosanoids

We evaluated the effects of 17α-E2 treatment on circulating eicosanoids (*Supplementary file 3*). Plasma eicosanoid analyses were performed by the UCSD Lipidomics Core as described previously (*Quehenberger et al., 2010*). Eicosanoids were isolated from plasma, extracted, separated using liquid chromatography, and analyzed with mass spectrometry (MDS SCIEX 4000 Q Trap; Applied Biosystems, Foster City, CA) (*Quehenberger et al., 2010*).

## Real-time PCR

To evaluate alterations in gene expression following 17α-E2 treatment, we performed qPCR for genes related to fibrosis, lipid metabolism, insulin resistance, and glucose metabolism in the liver. Total RNA was extracted using Trizol (Life Technologies, Carlsbad, CA) and was reverse transcribed to cDNA with the High-Capacity cDNA Reverse Transcription kit (Applied Biosystems, Foster City,

CA). Real-time PCR was performed in a QuantStudio 12K Flex Real Time PCR System (Thermofisher Scientific, Waltham, MA) using TaqMan Gene Expression Master Mix (Applied Biosystems/Thermofisher Scientific, Waltham, MA) and predesigned gene expression assays with FAM probes from Integrated DNA Technologies (Skokie, Illinois). Target gene expression was expressed as $2^{-\Delta\Delta CT}$ by the comparative CT method (*Livak and Schmittgen, 2001*) and normalized to the expression of TATA-Box Binding Protein (TBP) in liver.

## Western blotting

To determine if 17α-E2 altered hepatic insulin sensitivity, we evaluated phosphorylation status of AKT and FOXO1 following an insulin bolus. Liver was homogenized in RIPA Buffer (Cell Signaling, Danvers, MA) with protease and phosphatase inhibitors (Boston Bioproducts, Boston, MA). Total protein was quantified using BCA Protein Assay Reagent Kit (Pierce, Rockford, IL). Proteins were separated on an Any kD Criterion TGX Stain-Free Protein Gel (Biorad, Hercules, CA) at 75V for 150 min in Running Buffer (Cell Signaling, Danvers, MA) and transferred to a 0.2 μm pore size nitrocellulose membrane, (Biorad, Hercules, CA) at 75V for 90 min on ice. Primary antibodies used were pS256 FOX01 (Abcam ab131339, 1:1000), FOX01a (Abcam ab52857, 1:1000), pS473 AKT (Abcam ab81283, 1:3000), pan-AKT (Abcam ab179463, 1:10000), GAPDH (Abcam ab9485, 1:2500). Primary antibody detection was performed with IRDye 800CW Infrared Rabbit (LI-COR Biotechnology, Lincoln, NE) at 1:15,000 concentration. GAPDH was diluted in 5% dry milk (Cell Signaling, Danvers, MA), all other antibodies were diluted in 5% BSA (Cell Signaling, Danvers, MA). Blot imaging was done on Odyssey Fc Imaging System (LI-COR Biotechnology, Lincoln, NE) with a two-minute exposure time at 800λ, and protein detection and quantification were performed using Image Studio Software (LI-COR Biotechnology, Lincoln, NE).

## Statistical analyses

Results are presented as mean ± SEM unless otherwise stated with *p* values less than 0.05 considered to be significant unless otherwise specified. Analyses of differences between groups were performed by two-way ANOVA, two-way repeated measures ANOVA, or Student's t-test where appropriate using SigmaPlot 12.5 Software. A Benjamini-Hochberg multiple testing correction was applied to the F test result to correct for the number of transcripts, proteins, and fatty acids analyzed.

## Acknowledgements

We thank Dr. Kenneth Korach at the National Institute of Environmental Health Sciences for providing ERα KO and WT littermate mice. We also thank Dr. Lora Bailey-Downs, Richard Brush, and Michael Sullivan for technical support. This work was supported by the National Institutes of Health (R00 AG51661 and R01 AG069742 to MBS, T32 AG052363 to SNM, R01 EY030513 to M-PA, and R01 AG059430 to WMF), Veterans Affairs (I01B × 003906 to WMF) and pilot research funding from the Harold Hamm Diabetes Center (MBS and SNM), Einstein Nathan Shock Center (P30 AG038072) of Excellence in the Basic Biology of Aging (MBS), and OUHSC Lipidomics Core (P30 EY012190).

## Additional information

### Funding

| Funder | Grant reference number | Author |
|---|---|---|
| National Institutes of Health | R00 AG51661 | Michael B Stout |
| Harold Hamm Diabetes Center | Pilot Research Funding | Shivani N Mann Michael B Stout |
| National Institutes of Health | R01 AG069742 | Michael B Stout |
| National Institutes of Health | R01 AG059430 | Willard M Freeman |
| Veterans Affairs Oklahoma City | I01BX003906 | Willard M Freeman |
| University of Oklahoma Health | P30 EY012190 | Martin-Paul Agbaga |

| | | |
|---|---|---|
| Sciences Center | | |
| National Institutes of Health | R01 EY030513 | Martin-Paul Agbaga |
| National Institutes of Health | T32 AG052363 | Shivani N Mann |
| Einstein Nathan Shock Center | P30 AG038072 | Michael B Stout |

The funders had no role in study design, data collection and interpretation, or the decision to submit the work for publication.

## Author contributions

Shivani N Mann, Conceptualization, Formal analysis, Funding acquisition, Investigation, Methodology, Writing - original draft, Project administration, Writing - review and editing; Niran Hadad, Software, Formal analysis, Validation, Investigation, Visualization, Methodology, Writing - review and editing; Molly Nelson Holte, Formal analysis, Validation, Methodology; Alicia R Rothman, Formal analysis, Validation, Writing - review and editing; Roshini Sathiaseelan, Formal analysis, Investigation, Methodology; Samim Ali Mondal, Formal analysis, Investigation, Writing - review and editing; Martin-Paul Agbaga, Archana Unnikrishnan, Formal analysis, Investigation, Methodology, Writing - review and editing; Malayannan Subramaniam, Formal analysis, Supervision, Investigation, Methodology, Writing - review and editing; John Hawse, Formal analysis, Supervision, Validation, Investigation, Methodology, Writing - review and editing; Derek M Huffman, Data curation, Formal analysis, Validation, Investigation, Visualization, Methodology, Writing - review and editing; Willard M Freeman, Data curation, Software, Formal analysis, Supervision, Investigation, Visualization, Writing - review and editing; Michael B Stout, Conceptualization, Resources, Data curation, Formal analysis, Supervision, Funding acquisition, Validation, Investigation, Visualization, Methodology, Writing - original draft, Project administration, Writing - review and editing

## Author ORCIDs

Willard M Freeman ⓘ http://orcid.org/0000-0001-7027-999X
Michael B Stout ⓘ https://orcid.org/0000-0002-9996-9123

## Ethics

Animal experimentation: This study was performed in strict accordance with the recommendations in the Guide for the Care and Use of Laboratory Animals of the National Institutes of Health. All of the animals were handled according to approved institutional animal care and use committee (IACUC) protocols (#19-063-SEAHI) of the University of Oklahoma Health Science Center.

## Decision letter and Author response

Decision letter https://doi.org/10.7554/eLife.59616.sa1
Author response https://doi.org/10.7554/eLife.59616.sa2

# Additional files

## Supplementary files

• Supplementary file 1. Pairwise statistical comparisons of ERα binding. Negative binomial regression Wald post-hoc comparison test, FDR < 0.05. n = 3/group.

• Supplementary file 2. ERα binding motif analysis. Motif analysis was performed using HOMER with standard settings with the significance threshold set to FDR corrected p<0.05. Peak regions called for each treatment group were analyzed to identify enriched motifs relative to the entire genome. Hypergeometric test was used to test enrichment. Only motifs with FDR corrected p<0.05 were reported as significant. For pairwise differential motif enrichment or depletion across experimental groups, we utilized the hypergeometric test by using the number of sequences with motif from each group and total number of peaks as total sample size. Motifs that appear in less than five sequences between both test groups were removed. Benjamini-Hochberg multiple testing correction was utilized to control for false discovery rate (FDR < 0.05).

• Supplementary file 3. Circulating eicosanoid levels (pmol/ml). 17α-E2 mildly alters the circulating eicosanoid profile in obese middle-aged male mice. WT mice were provided 45% HFD (TestDiet 58V8)±17α-E2 (14.4ppm) for 14 weeks. Age-matched, male WT, chow-fed (TestDiet 58YP) mice were also evaluated as a normal-weight reference group. All data are presented as mean ± SEM and were analyzed by Student's t-test with the WT Chow group being excluded from statistical comparisons. n = 5–7.

• Transparent reporting form

### Data availability

Sequencing data has been deposited in GEO under accession code GSE151039.

The following dataset was generated:

| Author(s) | Year | Dataset title | Dataset URL | Database and Identifier |
|---|---|---|---|---|
| Stout M, Hawse J, Freeman W, Hadad N, Mann S | 2020 | Assessment of transcriptional ERa activity following exposure to 17a-E2 and 17b-E2 | https://www.ncbi.nlm.nih.gov/geo/query/acc.cgi?acc=GSE151039 | NCBI Gene Expression Omnibus, GSE151039 |

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
