## [Decision Letter]

**Acceptance summary:**

This manuscript expands upon and provides an important mechanistic explanation for previous published studies reporting that treatment with 17α-estradiol (17α-E2), extends lifespan in male mice and also reverses the harmful metabolic changes associated with obesity. The authors confirm through genomic and knock out studies that this effect is dependent upon the presence of the estrogen receptor a rather than a novel receptor as was previously speculated. Moreover, they elucidate a potential mechanism that contributes to the longevity extending effects of 17α-E2 by life extending effects primarily to modulating hypothalamic and hepatic glucose metabolism.

**Decision letter after peer review:**

Thank you for submitting your article "Health benefits attributed to 17α-estradiol, a lifespan-extending compound, are mediated through estrogen receptor α" for consideration by *eLife*. Your article has been reviewed by four peer reviewers, one of whom is a member of our Board of Reviewing Editors, and the evaluation has been overseen Jessica Tyler as the Senior Editor. The following individual involved in review of your submission has agreed to reveal their identity: Jacob C Kimmel (Reviewer #3).

The reviewers have discussed the reviews with one another and the Reviewing Editor has drafted this decision to help you prepare a revised submission.

Summary:

This work has the potential to make an important original contribution to the aging biology literature and includes interesting new findings regarding the role of 17α-E2 in lifespan extension. The authors provide persuasive evidence of the role of classic estradiol receptor ERα in 17α-E2 signaling and mediating the metabolic effects of 17α-E2. However, far reaching and unsupported claims are made regarding the central tenets of this manuscript as is stated in their Abstract – namely sex-specific differences and that of tissue specific mediation of 17α-E2 effects facilitating the therapeutic benefit. While the tissue specific data are suggestive that the liver and hypothalamus facilitate the beneficial effects of 17α-E2, these data do not appear to have been appropriately statistically analyzed. These need to be addressed prior to publication.

The authors need to provide additional evidence for the tissue-specific nature of 17α-E2s effects on metabolism and lifespan, as well as correct/redo/undertake their statistical analyses using appropriate methods. Moreover, the experimental design needs to be more clearly documented and the data presented in a legible and interpretable manner. As such, the manuscript requires a complete reanalysis of current data and a complete rewrite, before we could consider its publication in *eLife*.

Essential revisions:

1) The experimental design needs to be more clearly documented, and the data presented in legible figures with detailed legends that alert the reader to the salient methods and findings. The authors should include a brief rational for each of the experiments used and their choice of cells and provide a concise description of the methods used, and sample size. The authors need to articulate how the significance for RNA-seq and ChIP-seq data was assessed.

2) If the authors already have the complementary data on female mice on a high fat diet (rather than on normal chow) to facilitate a direct comparison with the male mice on a high fat diet and, thereby, support their sex-specific claims, it would strengthen the paper considerably, if not their claims on sex specificity should be removed from this paper as one cannot directly compare females on normal chow with males on a HFD.

3) The authors state that their ChIP-seq data reveal nearly identical ERα binding patterns with17α-E2 and 17β-E2, however these have not been rigorously analyzed. The authors need to undertake actual statistical comparisons across groups rather than rely solely on qualitative assessments.

4) The authors need to provide additional evidence for the tissue-specific nature of 17α-E2s effects on metabolism and lifespan. While they present convincing data that administration of 17α-E2 has direct effects on liver and hypothalamus, they have not provided definitive evidence that these tissues are directly responsible for the beneficial effects on metabolism and lifespan. The Abstract and Results section should be appropriately tempered.

5) The authors claim that 17α-E2 reverses cellular senescence in the liver but provide limited conclusive supporting data. The authors need to provide additional evidence to support these claims.

6) The motif comparison for the ChIP-seq data in Figure 1B is not described sufficiently to allow for evaluation by readers. Moreover, the authors do not appear to have employed appropriate statistical analyses. Please describe the statistical tests employed.

[Editors' note: further revisions were suggested prior to acceptance, as described below.]

Thank you for resubmitting your article "Health benefits attributed to 17α-estradiol, a lifespan-extending compound, are mediated through estrogen receptor α" for consideration by *eLife*. Your article has been reviewed by Jessica Tyler as the Senior Editor, a Reviewing Editor, and three reviewers. The following individual involved in review of your submission this round has agreed to reveal their identity: Rozalyn Anderson (Reviewer #6).

The reviewers have discussed the reviews with one another and the Reviewing Editor has drafted this decision to help you prepare a revised submission. Overall, all agree the manuscript has been vastly improved, however several key concerns were ignored in your revision and still need to be addressed before this article can be accepted for publication.

The genomics analyses remain difficult to interpret and the prior request for clearer, more interpretable figures was ignored. Please redo these figures, as per the comments from the first set of reviewers.

Moreover, the Materials and methods are insufficient for reproduction, and some relevant statistical tests are omitted. These statistical information should be included within the body of the manuscript adjacent to the specific results in which they were used.

The female data remain problematic and while there are understandable issues with KO females; given the differing experimental designs and the fact that the bulk of the data are exclusively in males, we believe that the paper would be considerably stronger if these inconclusive data on females given their different treatments/diet were excluded from the paper and the Abstract as well. as the paper modified accordingly. We are not asking for additional data.

The text should be tightened up, removing speculative conclusions not based. upon the data presented. The manuscript and in particular the Abstract still contains unsupported claims that 17α-E2 acts through liver and hypothalamus, despite the authors pointing out that this claim is "speculation" later in the Discussion.

Below please find the reviewer's comments.

Reviewer #3:

The authors have made considerable improvements to the original manuscript. However, some concerns remain that should be addressed. While the descriptions are improved, the authors' genomics analyses remain difficult to interpret from the unchanged figures, the Materials and methods are insufficient for reproduction, and some relevant statistical tests are omitted. The authors have appropriately moderated their claims of tissue-specific mediators of 17-E2 benefit, but the original, unqualified claims nonetheless remain in the Abstract.

Essential revisions:

1) The p-values presented in Figure 1B are still much smaller than those present in the supplementary data (Supplementary file 1). For instance, the first motif listed in Figure 1B shows p-values at log10(p) = -5000, but the lowest in the table is ~-2000. What accounts for this discrepancy?

2) The claim that there are "no differences in the binding motifs identified between treatment groups" suggests a statistical comparison was performed across all pairwise combinations of treatment groups. Is this the case? If so, the tests should be specified in the Figure legend and explicitly described in the Materials and methods (e.g. We performed pairwise comparisons between treatment groups…). It also seems plausible that there may be differences in the enrichment of ERα among its motifs depending on the ligand. Rather than making binary calls for motif identification/non-identification, the authors should compare the enrichment levels across treatment groups (hypergeometric would again be appropriate).

3) The manuscript still contains prominent claims that 17α-E2 acts through liver and hypothalamus, despite the authors pointing out that this claim is "speculation" later in the manuscript. The Abstract contains "that 17α-E2 acts primarily through the liver and hypothalamus to improve metabolic parameters" while the later Introduction states "we speculate that 17α-E2 acts through ERα in the liver and/or hypothalamus to improve metabolic homeostasis in male mammals" and the Results state "…suggests that the liver and hypothalamus are two primary sites of action for the regulation of metabolic parameters by 17α-E2." The authors should adopt the latter qualifications for all such statements. As agreed in their response to reviewers, they do not present evidence to support the former, stronger claim that remains prominently in the Abstract.

Reviewer #6:

The manuscript is much improved, the central message is certainly of interest and the genetic component adds pretty convincing evidence that the 17-α-E2 is indeed working through or at least requires the ERα receptor to improve metabolic indices in male mice on a high fat diet.

The authors have been very responsive to the concerns raised – I think it would be no harm at all to name the statistical tests used in the body of the paper not just to have them better described in the Materials and methods which they have done as per the Editor's suggestion.

The Discussion is a little redundant in places with the Result section – the authors might want to trim out the repetition so that the major points of Discussion, context, and interpretation are not lost to the reader.

Reviewer #7:

Authors have nicely addressed most of the previous concerns.

The Female data are still problematic. Due to the differing design, interpretation is unknown whether difference is due to females, due to diet, etc. Understandable there are issues with the KO female, but suggest perhaps removing these data. Particularly since the remainder of data are in males exclusively.

---

## [Author Response]

Essential revisions:1) The experimental design needs to be more clearly documented, and the data presented in legible figures with detailed legends that alert the reader to the salient methods and findings. The authors should include a brief rational for each of the experiments used and their choice of cells and provide a concise description of the methods used, and sample size. The authors need to articulate how the significance for RNA-seq and ChIP-seq data was assessed.

We appreciate these suggestions and apologize for our initial submission lacking appropriate detail. In response to these requests, we:

a) completely reanalyzed all of the data and rewrote several sections of the manuscript to include more detail related to the experimental design and statistical approaches employed

b) significantly expanded the figure legends by including more detail related to approach and analyses

c) added brief rationales to the Results section and Materials and methods section for each experiment undertaken

d) significantly expanded methodological and statistical sections describing how the RNA-seq and ChIP-seq experiments were performed and analysed.

2) If the authors already have the complementary data on female mice on a high fat diet (rather than on normal chow) to facilitate a direct comparison with the male mice on a high fat diet and, thereby, support their sex-specific claims, it would strengthen the paper considerably, if not their claims on sex specificity should be removed from this paper as one cannot directly compare females on normal chow with males on a HFD.

We understand that the decision to place male mice on high-fat diets and female mice on chow diets could be confusing, so we provided additional background on the ERα KO model and a rationale for why this was done. As is now stated in the manuscript (Discussion), “female mice present a greater phenotypic response than males to the ablation of ERα, thereby exacerbating obesity and metabolic dysfunction which makes comparisons to female WT mice as well as their male littermates problematic. For this reason, we chose not to provide HFD to female mice in these studies.” In addition to providing more detail about our rationale, we have also tempered our conclusions throughout the manuscript related to sexually dimorphic effects of 17α-E2 in our studies.<bold />*3) The authors state that their ChIP-seq data reveal nearly identical ERα binding patterns with 17α-E2 and 17β-E2, however these have not been rigorously analyzed. The authors need to undertake actual statistical comparisons across groups rather than rely solely on qualitative assessments.*

We apologize that the statistical analyses we used to compare the ChIP-seq data were poorly described in our initial submission. We have now significantly expanded the methodological and statistical sections describing how the ChIP-seq experiments were performed and analyzed.

4) The authors need to provide additional evidence for the tissue-specific nature of 17α-E2s effects on metabolism and lifespan. While they present convincing data that administration of 17α-E2 has direct effects on liver and hypothalamus, they have not provided definitive evidence that these tissues are directly responsible for the beneficial effects on metabolism and lifespan. The Abstract and Results section should be appropriately tempered.

We appreciate this concern and have tempered our conclusions within the manuscript to convey that our data are highly suggestive of tissue-specific effects of 17α-E2 actions, but that additional studies will be needed to definitively prove this. In the final paragraph of the Discussion we state that, “Future studies will be needed to confirm that 17α-E2 acts predominantly through ERα in a cell-type-specific manner in the liver and hypothalamus to modulate systemic metabolic homeostasis”.

5) The authors claim that 17α-E2 reverses cellular senescence in the liver but provide limited conclusive supporting data. The authors need to provide additional evidence to support these claims.

We agree that our data does not allow for any definitive conclusions to be drawn regarding how 17α-E2 modulates cellular senescence in the liver, therefore we removed any reference to this in the text. This portion of the manuscript and Figure 5 now focuses exclusively on markers of liver fibrosis and insulin sensitivity.

6) The motif comparison for the ChIP-seq data in Figure 1B is not described sufficiently to allow for evaluation by readers. Moreover, the authors do not appear to have employed appropriate statistical analyses. Please describe the statistical tests employed.

We apologize that the statistical analyses we used to analyze the ChIP-seq data were poorly described in our initial submission. We have significantly expanded the methodological and statistical sections describing how the ChIP-seq and motif enrichment analyses were performed and analyzed, and additional citations were also added.

[Editors' note: further revisions were suggested prior to acceptance, as described below.]

Reviewer #3:The authors have made considerable improvements to the original manuscript. However, some concerns remain that should be addressed. While the descriptions are improved, the authors' genomics analyses remain difficult to interpret from the unchanged figures, the Materials and methods are insufficient for reproduction, and some relevant statistical tests are omitted. The authors have appropriately moderated their claims of tissue-specific mediators of 17-E2 benefit, but the original, unqualified claims nonetheless remain in the Abstract.

We thank reviewer #3 for their suggestions. We have addressed all concerns in a detailed fashion below.<bold />*Essential revisions:*

1) The p-values presented in Figure 1B are still much smaller than those present in the supplementary data (Supplementary file 1). For instance, the first motif listed in Figure 1B shows p-values at log10(p) = -5000, but the lowest in the table is ~-2000. What accounts for this discrepancy?

We apologize for the confusion related to this. In the revised Figure 1C the log p-value is presented which corresponds to column E in Supplementary file 2. The former Supplementary file 1 is now Supplementary file 2 because we added an additional Supplementary file that corresponds to the new Figure 1B.<bold />*2) The claim that there are "no differences in the binding motifs identified between treatment groups" suggests a statistical comparison was performed across all pairwise combinations of treatment groups. Is this the case? If so, the tests should be specified in the Figure legend and explicitly described in the Materials and methods (e.g. We performed pairwise comparisons between treatment groups…). It also seems plausible that there may be differences in the enrichment of ERα among its motifs depending on the ligand. Rather than making binary calls for motif identification/non-identification, the authors should compare the enrichment levels across treatment groups (hypergeometric would again be appropriate).*

The presentation of the ERα ChIP-seq data has been extensively revised. First, the Figure 1 panels have been updated to clarify axes and labels. Second, a new panel (Figure 1B) has been added to more clearly demonstrate that ERα binds to the same genomic locations and to similar magnitudes with both 17α-E2 and 17β-E2 exposure. We have also clarified our findings in the Results, Materials and methods, and figure legends while noting the specific statistical tests used. Regarding the question related to differential motif enrichment, there were no differences between 17α-E2 and 17β-E2 in the motif enrichment, which is aligned with the observation that ERα is binding to the same genomic locations following exposure to both ligands.

3) The manuscript still contains prominent claims that 17α-E2 acts through liver and hypothalamus, despite the authors pointing out that this claim is "speculation" later in the manuscript. The Abstract contains "that 17α-E2 acts primarily through the liver and hypothalamus to improve metabolic parameters" while the later Introduction states "we speculate that 17α-E2 acts through ERα in the liver and/or hypothalamus to improve metabolic homeostasis in male mammals" and the Results state "…suggests that the liver and hypothalamus are two primary sites of action for the regulation of metabolic parameters by 17α-E2." The authors should adopt the latter qualifications for all such statements. As agreed in their response to reviewers, they do not present evidence to support the former, stronger claim that remains prominently in the Abstract.

We appreciate this astute observation by the reviewer and have updated the Abstract to reflect the tempered conclusions in other parts of manuscript.

Reviewer #6:The manuscript is much improved, the central message is certainly of interest and the genetic component adds pretty convincing evidence that the 17-α-E2 is indeed working through or at least requires the ERα receptor to improve metabolic indices in male mice on a high fat diet.

We thank reviewer #6 for their kind words and suggestions. We have addressed all concerns in a detailed fashion below.<bold />

The authors have been very responsive to the concerns raised – I think it would be no harm at all to name the statistical tests used in the body of the paper not just to have them better described in the Materials and methods which they have done as per the Editor's suggestion.

We have expanded the details on the statistical tests used in the Results, text, figure legends, and Materials and methods.

The Discussion is a little redundant in places with the result section – the authors might want to trim out the repetition so that the major points of discussion, context, and interpretation are not lost to the reader.

We appreciate this suggestion and have further refined the Discussion.

Reviewer #7:Authors have nicely addressed most of the previous concerns.

We thank reviewer #7 for their kind words and suggestions. We have addressed the concerns related to the inclusion of female mice below.<bold /><italic />

The Female data are still problematic. Due to the differing design, interpretation is unknown whether difference is due to females, due to diet, etc. Understandable there are issues with the KO female, but suggest perhaps removing these data. Particularly since the remainder of data are in males exclusively.

We appreciate this suggestion, but feel the female data is sufficiently interpretable and supports other publications in the field. Additionally, since our previous resubmission to *eLife* we have published a report demonstrating that OVX in females engenders 17α-E2 responsiveness, which supports our contention that endogenous 17β-E2 plays a key role in rendering female mice unresponsive to exogenous 17α-E2 treatment. This new publication (Mann et al., 2020), coupled with the genomic and transcriptional data presented in the current report (Figure 1, Figure 1—figure supplement 1, Supplementary file 1, Supplementary file 2), allows for clearer interpretation of the female data from the current set of studies. Of particular relevance, we are not the first group to report that 17α-E2 has limited effectiveness on a variety of variables in intact female mice (see Strong et al., 2016, Harrison et al., 2014, Garratt et al., 2017, PMID: 30740872); therefore, our female data presented herein that demonstrates a lack of effects of 17α-E2 on metabolic variables supports previous reports. Lastly, we also explicitly describe within the manuscript our rationale for not subjecting female mice to high-fat feeding (the ERαKO naturally develops obesity) and discuss the importance of future studies evaluating the effects of 17α-E2 in female mice following diet-induced obesity (HFD), which we openly acknowledge was not done in the current set of experiments. In an effort to find common ground, we have further tempered our conclusions related to our observations in female mice in these studies.